# Rebooting ACGAN: Auxiliary Classifier GANs with Stable Training

**Minguk Kang**    **Woohyeon Shim**    **Minsu Cho**    **Jaesik Park**

Pohang University of Science and Technology (POSTECH), South Korea
`{mgkang, wh.shim, mscho, jaesik.park}@postech.ac.kr`

## Abstract

Conditional Generative Adversarial Networks (cGAN) generate realistic images by incorporating class information into GAN. While one of the most popular cGANs is an auxiliary classifier GAN with softmax cross-entropy loss (ACGAN), it is widely known that training ACGAN is challenging as the number of classes in the dataset increases. ACGAN also tends to generate easily classifiable samples with a lack of diversity. In this paper, we introduce two cures for ACGAN. First, we identify that gradient exploding in the classifier can cause an undesirable collapse in early training, and projecting input vectors onto a unit hypersphere can resolve the problem. Second, we propose the Data-to-Data Cross-Entropy loss (D2D-CE) to exploit relational information in the class-labeled dataset. On this foundation, we propose the Rebooted Auxiliary Classifier Generative Adversarial Network (ReACGAN). The experimental results show that ReACGAN achieves state-of-the-art generation results on CIFAR10, Tiny-ImageNet, CUB200, and ImageNet datasets. We also verify that ReACGAN benefits from differentiable augmentations and that D2D-CE harmonizes with StyleGAN2 architecture. Model weights and a software package that provides implementations of representative cGANs and all experiments in our paper are available at https://github.com/POSTECH-CVLab/PyTorch-StudioGAN.

## 1 Introduction

Generative Adversarial Networks (GAN) [1] are known for the forefront approach to generating high-fidelity images of diverse categories [2, 3, 4, 5, 6, 7, 8, 9]. Behind the sensational generation ability of GANs, there has been tremendous effort to develop adversarial objectives free from the vanishing gradient problem [10, 11, 12], regularizations for stabilizing adversarial training [11, 13, 14, 3, 15, 16, 17], and conditioning techniques to support the adversarial training using category information of the dataset [18, 19, 20, 21, 22, 23, 24, 25, 26, 27, 28]. Subsequently, the conditioning techniques have become the *de facto* standard for high-quality image generation. The models with the conditioning methods are called conditional Generative Adversarial Networks (cGAN), and cGANs can be divided into two groups depending on the discriminator's conditioning way: classifier-based GANs [18, 20, 23, 25, 27] and projection-based GANs [19, 3, 4, 28].

The classifier-based GANs facilitate an auxiliary classifier to generate class-specific images by penalizing the generator if the synthesized images are not consistent with the conditioned labels. ACGAN [18] has been one of the widely used classifier-based GANs for its simple design and satisfactory generation performance. While ACGAN can exploit class information by pushing and pulling classifier's weights (proxies) against image embeddings [23], it is well known that ACGAN training is prone to collapsing at the early stage of training as the number of classes increases [19, 23, 25, 27]. In addition, the generator of ACGAN tends to generate easily classifiable images at the cost of reduced diversity [18, 19, 27]. Projection-based GANs, on the other hand, have shown cutting-edge generation results on datasets with a large number of categories. SNGAN [3],

35th Conference on Neural Information Processing Systems (NeurIPS 2021).

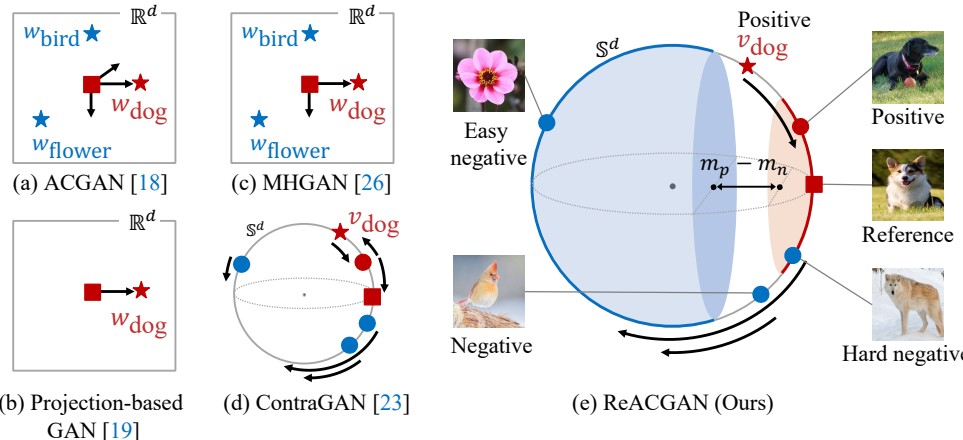

(a) ACGAN [18]  (c) MHGAN [26]

(b) Projection-based GAN [19]  (d) ContraGAN [23]  (e) ReACGAN (Ours)

Figure 1: Schematics that depict how cGANs perform conditioning. The red color means positive samples/proxy, and the blue color indicates negative samples/proxies. Arrows represent push-pull forces based on the reference sample. The length of an arrow indicates the magnitude of the force.

SAGAN [29], and BigGAN [4] are representatives in this family and can generate realistic images on CIFAR10 [30] and ImageNet [31] datasets. However, projection-based GANs only consider a pairwise relationship between an image and its label proxy (data-to-class relationships). As the result, the projection-based GANs can miss an additional opportunity to consider relation information between data instances (data-to-data relationships) as discovered by [23].

In this paper, we analyze why ACGAN training becomes unstable as the number of classes increases and propose remedies for (1) the instability and (2) the relatively poor generation performance of ACGAN compared with the projection-based models. First, we begin by analytically deriving the gradient of the softmax cross-entropy loss used in ACGAN. By examining the exact values of analytic gradients, we discover that the unboundedness of input feature vectors and poor classification performance in the early training stage can cause an undesirable gradient exploding problem. Second, with alleviating the instability, we propose the Rebooted Auxiliary Classifier Generative Adversarial Networks (ReACGAN) using the Data-to-Data Cross-Entropy loss (D2D-CE). ReACGAN projects image embeddings and proxies onto a unit hypersphere and computes similarities for data-to-data and data-to-class consideration. Additionally, we introduce two margin values for intra-class variations and inter-class separability. In this way, ReACGAN overcomes the training instability and can exploit additional supervisory signals by explicitly considering data-to-class and data-to-data relationships, and also by implicitly looking at class-to-class relationships in the same mini-batch.

To validate our model, we conduct image generation experiments on CIFAR10 [30], Tiny-ImageNet [32], CUB200 [33], and ImageNet [31] datasets. Through extensive experiments, we demonstrate that ReACGAN beats both the classifier-based and projection-based GANs, improving over the state of the art by 2.5%, 15.8%, 5.1%, and 14.5% in terms of Fréchet Inception Distance (FID) [34] on the four datasets, respectively. We also verify that ReACGAN benefits from consistency regularization [16] and differentiable augmentations [9, 8] for limited data training. Finally, we confirm that D2D-CE harmonizes with the StyleGAN2 architecture [7].

## 2    Background: Generative Adversarial Networks

Generative Adversarial Network (GAN) [1] is an implicit generative model that aims to generate a sample indistinguishable from the real. GAN consists of two networks: a *Generator* $G : \mathcal{Z} \longrightarrow \mathcal{X}$ that tries to map a latent variable $\mathbf{z} \sim p(\mathbf{z})$ into the real data space $\mathcal{X}$ and a *Discriminator* $D : \mathcal{X} \longrightarrow [0, 1]$ that strives to discriminate whether a given sample $\mathbf{x}$ is from the real data distribution $p_{\text{real}}(\mathbf{x})$ or from the implicit distribution $p_{\text{gen}}(G(\mathbf{z}))$ derived from the generator $G(\mathbf{z})$. The objective of a vanilla GAN [1] can be expressed as follows:

$$\min_{G} \max_{D} \; \mathbb{E}_{\mathbf{x} \sim p(\mathbf{x})}[\log(D(\mathbf{x}))] + \mathbb{E}_{\mathbf{z} \sim p(\mathbf{z})}[\log(1 - D(G(\mathbf{z})))]. \qquad (1)$$

While GANs have shown impressive results in the image generation task [2, 35, 11, 13], training GANs often ends up encountering a mode-collapse problem [36, 11, 12]. As one of the prescriptions for stabilizing and reinforcing GANs, training GANs with categorical information, named conditional Generative Adversarial Networks (cGAN), is suggested [37, 18, 19]. Depending on the presence of explicit classification losses, cGAN can be divided into two groups: Classifier-based GANs [18, 20, 23, 25, 27] and Projection-based GANs [19, 3, 4, 28]. One of the widely used classifier-based GANs is ACGAN [18], and ACGAN utilizes softmax cross-entropy loss to perform classification task with adversarial training. Although ACGAN has shown satisfactory generation results, training ACGAN becomes unstable as the number of classes in the training dataset increases [19, 23, 25, 27]. Besides, ACGAN tends to generate easily classifiable images at the cost of limited diversity [18, 19, 27]. To alleviate those problems, Zhou *et al.* [38] have proposed performing adversarial training on the classifier. Gong *et al.* [20] have introduced an additional classifier to eliminate a conditional entropy minimization process in the adversarial training. However, ACGAN training still suffers from the early-training collapse issue and the reduced diversity problem when trained on datasets with a large number of class categories, such as Tiny-ImageNet [32] and ImageNet [31].

In these circumstances, Miyato *et al.* [19] have proposed a projection discriminator for cGANs and have shown significant improvement in generating the ImageNet dataset. Motivated by the promising result of the projection discriminator, many projection-based GANs [3, 29, 4, 16, 6, 17] have been proposed and become the standard for conditional image generation. In this paper, we revisit ACGAN and unveil why ACGAN training is so unstable. Coping with the instability, we propose the Rebooted Auxiliary Classifier GANs (ReACGAN) for high-quality and diverse image generation.

## 3    Rebooting Auxiliary Classifier GANs

### 3.1    Feature Normalization

To uncover a nuisance that can cause the instability of ACGAN, we start by analytically deriving the gradients of weight vectors in the softmax classifier. Let the part of the discriminator before the fully connected layer be a *Feature extractor* $F : \mathcal{X} \longrightarrow \mathcal{F} \in \mathbb{R}^d$ and let *Classifier* $C : \mathcal{F} \longrightarrow \mathbb{R}^c$ be a single fully connected layer parameterized by $\boldsymbol{W} = [\boldsymbol{w}_1, ..., \boldsymbol{w}_c] \in \mathbb{R}^{d \times c}$, where $c$ denotes the number of classess. We sample training images $\boldsymbol{X} = \{\boldsymbol{x}_1, ..., \boldsymbol{x}_N\}$ and integer labels $\boldsymbol{y} = \{y_1, ..., y_N\}$ from the joint distribution $p(\mathbf{x}, \mathbf{y})$. Using the notations above, we can express the empirical cross-entropy loss used in ACGAN [18] as follows:

$$\mathcal{L}_{\text{CE}} = -\frac{1}{N} \sum_{i=1}^{N} \log \left( \frac{\exp\left(F(\boldsymbol{x}_i)^\top \boldsymbol{w}_{y_i}\right)}{\sum_{j=1}^{c} \exp\left(F(\boldsymbol{x}_i)^\top \boldsymbol{w}_j\right)} \right). \tag{3}$$

Based on Eq. (3), we can derive the derivative of the cross-entropy loss, w.r.t $\boldsymbol{w}_{k \in \{1, ..., c\}}$ as follows:

$$\frac{\partial \mathcal{L}_{\text{CE}}}{\partial \boldsymbol{w}_k} = -\frac{1}{N} \sum_{i=1}^{N} \left\{ F(\boldsymbol{x}_i) \left( \mathbf{1}_{y_i=k} - p_{i,k} \right) \right\}, \tag{4}$$

where $\mathbf{1}_{y_i=k}$ is an indicator function that will output 1 if $y_i = k$ is satisfied, and $p_{i,k}$ is a class probability that represents the probability that $i$-th sample belongs to class $k$, mathematically $\frac{\exp\left(F(\boldsymbol{x}_i)^\top \boldsymbol{w}_k\right)}{\sum_{j=1}^{c} \exp\left(F(\boldsymbol{x}_i)^\top \boldsymbol{w}_j\right)}$. The equation above implies that the norm of the gradient of the softmax cross-entropy loss is coupled with the norms and directions of each input feature map $F(\boldsymbol{x}_i)$ and the class probabilities. In the early training stage, the classifier is prone to making incorrect predictions, resulting in low probabilities. This phenomenon occurs more frequently as the number of categories in the dataset increases. As the result, the norm of the gradient $|\frac{\partial \mathcal{L}_{\text{CE}}}{\partial \boldsymbol{w}_k}|$ begins to explode as the vector $F(\boldsymbol{x}_i)$ stretches out to $\boldsymbol{w}_k$ direction but being located close to the the other vectors $\boldsymbol{w}_{j \in \{1, ..., c\} \setminus \{k\}}$. This often breaks the balance between adversarial learning and classifier training, leading to an early-training collapse. Once the early-training collapse occurs, ACGAN training concentrates on classifying categories of images instead of discriminating the authenticity of given samples. We experimentally demonstrate that the average norm of ACGAN's input feature maps increases as the training progresses (Fig. 2a). Accordingly, the average norm of the gradients increases sharply at the early training stage and decreases with the high class probabilities of the classifier (Fig. 2b, Fig. A3 in Appendix). While the average norm of gradients decreases at some point, the FID value of ACGAN does not decrease, implying the collapse of ACGAN training (Fig. 2c).

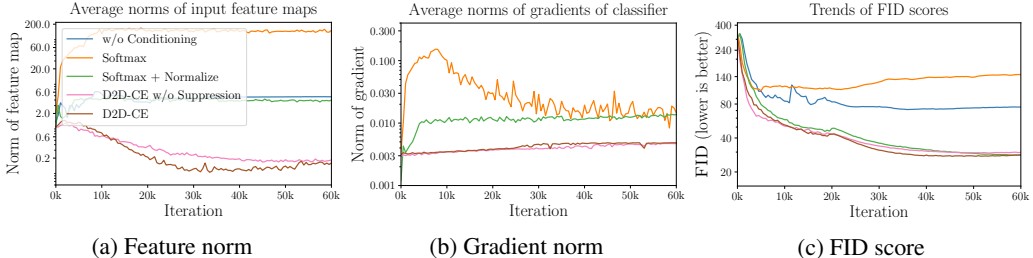

| | |
|---|---|
| (a) Feature norm | (b) Gradient norm |

(c) FID score

Figure 2: Merits of integrating feature normalization and data-to-data relationship consideration. All experiments are conducted on Tiny-ImageNet [32] dataset. (a) Average norms of input feature maps $F(\cdot)$, (b) average norms of gradients of classification losses, and (c) trends of FID scores. Compared with ACGAN [18], the proposed ReACGAN does not experience the training collapse problem caused by excessively large norms of feature maps and gradients at the early training stage. In addition, ReACGAN can converge to better equilibrium by considering data-to-data relationships with easy positive and negative sample suppression.

As one of the remedies for the gradient exploding problem, we find that simply *normalizing the feature embeddings onto a unit hypersphere* $\frac{F(\boldsymbol{x}_i)}{||F(\boldsymbol{x}_i)||}$ effectively resolves ACGAN's early-training collapse (see Fig. 2). The motivation is that normalizing the features onto the hypersphere makes the norms of feature maps equal to 1.0. Thus, the discriminator does not experience the gradient exploding problem. From the next section, we will deploy a linear projection layer $P$ on the feature extractor $F$. And, we will normalize both the embeddings from the projection layer and the weight vectors $(\boldsymbol{w}_1, ..., \boldsymbol{w}_c)$ in the classifier since normalizing both the embeddings and the weight vectors does not degrade image generation performance. We denote the normalized embedding $\frac{P(F(\boldsymbol{x}_i))}{||P(F(\boldsymbol{x}_i))||}$ as $\boldsymbol{f}_i$ and the normalized weight vector $\frac{\boldsymbol{w}_{y_i}}{||\boldsymbol{w}_{y_i}||}$ as $\boldsymbol{v}_{y_i}$.

### 3.2 Data-to-Data Cross-Entropy Loss (D2D-CE)

We expand the feature normalized softmax cross-entropy loss described in Sec. 3.1 to the Data-to-Data Cross-Entropy (D2D-CE) loss. The motivations are summarized into two points: (1) replacing data-to-class similarities in the denominator of Eq. (3) with data-to-data similarities and (2) introducing two margin values to the modified softmax cross-entropy loss. We expect that point (1) will encourage the feature extractor to consider data-to-class as well as data-to-data relationships, and that point (2) will guarantee inter-class separability and intra-class variations in the feature space while preventing ineffective gradient updates induced by easy negative and positive samples. To develop the feature normalized cross-entropy loss into D2D-CE, we replace the similarities between a sample embedding and all proxies except for the positive one in the denominator, $\sum_{j \in \{1,...,c\} \setminus \{y_i\}} \exp(\boldsymbol{f}_i^\top \boldsymbol{v}_j)$ with similarities between negative samples in the same mini-batch. The modified cross-entropy loss can be expressed as follows:

$$\mathcal{L}'_{\text{CE}} = -\frac{1}{N} \sum_{i=1}^{N} \log \left( \frac{\exp(\boldsymbol{f}_i^\top \boldsymbol{v}_{y_i}/\tau)}{\exp(\boldsymbol{f}_i^\top \boldsymbol{v}_{y_i}/\tau) + \sum_{j \in \mathcal{N}(i)} \exp(\boldsymbol{f}_i^\top \boldsymbol{f}_j/\tau)} \right), \tag{5}$$

where $\tau$ is a temperature, and $\mathcal{N}(i)$ is the set of indices that point locations of the negative samples whose labels are different from the reference label $\boldsymbol{v}_{y_i}$ in the mini-batch. The self-similarity matrix of samples in the mini-batch is used to calculate the similarities between negative samples $\boldsymbol{f}_i^\top \boldsymbol{f}_{j \in \mathcal{N}(i)}$ with a false negative mask (see Fig. 3). Thus, Eq. (5) enables the discriminator to contrastively compare visual differences between multiple images and can supply more informative supervision for image conditioning. Finally, we introduce two margin hyperparameters to $\mathcal{L}'_{\text{CE}}$ and name it *Data-to-Data Cross-Entropy loss* (D2D-CE). The proposed D2D-CE can be expressed as follows:

$$\mathcal{L}_{\text{D2D-CE}} = -\frac{1}{N} \sum_{i=1}^{N} \log \left( \frac{\exp([\boldsymbol{f}_i^\top \boldsymbol{v}_{y_i} - m_p]_-/\tau)}{\exp([\boldsymbol{f}_i^\top \boldsymbol{v}_{y_i} - m_p]_-/\tau) + \sum_{j \in \mathcal{N}(i)} \exp([\boldsymbol{f}_i^\top \boldsymbol{f}_j - m_n]_+/\tau)} \right), \tag{6}$$

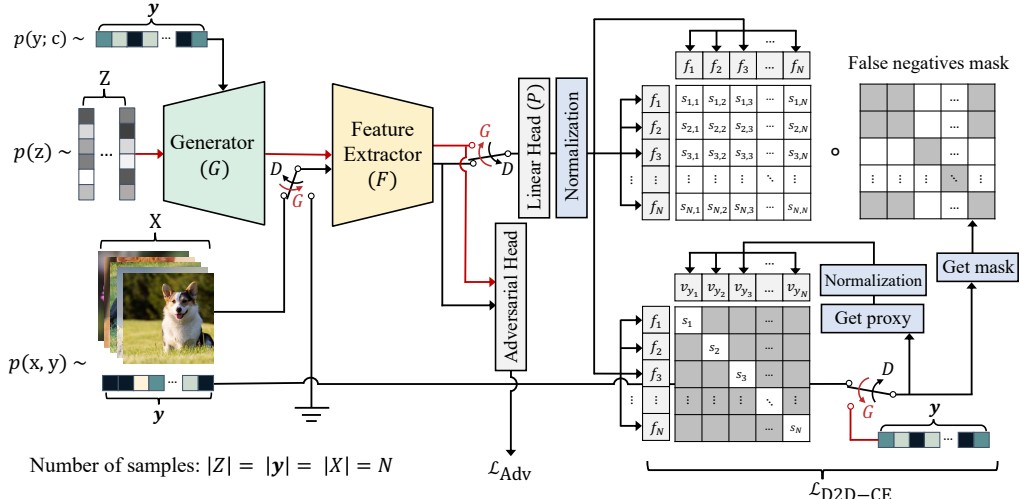

Number of samples: $|Z| = |\boldsymbol{y}| = |X| = N$

Figure 3: Overview of ReACGAN. ReACGAN performs adversarial training using the loss $\mathcal{L}_{\text{Adv}}$. At the same time, ReACGAN tries to minimize data-to-data cross-entropy loss (D2D-CE) on the linear head ($P$) to exploit relational information in the labeled dataset. ∘ means element-wise product. Note that *False negatives mask* operates on the similarity matrix between two batches of sample embeddings to compute the similarities between negative samples in the denominator of Eq. (6).

where $m_p$ is a margin for suppressing a high similarity value between a reference sample and its corresponding proxy (easy positive), $m_n$ is a margin for suppressing low similarity values between negatives samples (easy negatives). $[\cdot]_-$ and $[\cdot]_+$ denote $\min(\cdot, 0)$ and $\max(\cdot, 0)$ functions, respectively.

### 3.3 Useful Properties of Data-to-Data Cross-Entropy Loss

In this subsection, we explain four useful properties of D2D-CE. Let $s_q$ be a similarity between the normalized embedding $\boldsymbol{f}_q$ and the corresponding normalized proxy $\boldsymbol{v}_{y_q}$, $s_{q,r}$ be a similarity between $\boldsymbol{f}_q$ and $\boldsymbol{f}_r$, and $a$ and $b$ be arbitrary indices of negative samples, *i.e.*, $a, b \in \mathcal{N}(q)$. Then the properties of D2D-CE can be summarized as follows:

**Property 1.** *Hard negative mining. If the value of $s_{q,a}$ is greater than $s_{q,b}$, the derivative of $\mathcal{L}_{D2D\text{-}CE}$ w.r.t $s_{q,a}$ is greater than or equal to the derivative w.r.t $s_{q,b}$; that is $\frac{\partial \mathcal{L}_{D2D\text{-}CE}}{\partial s_{q,a}} \geq \frac{\partial \mathcal{L}_{D2D\text{-}CE}}{\partial s_{q,b}} \geq 0$.*

**Property 2.** *Easy positive suppression. If $s_q - m_p \geq 0$, the derivative of $\mathcal{L}_{D2D\text{-}CE}$ w.r.t $s_q$ is 0.*

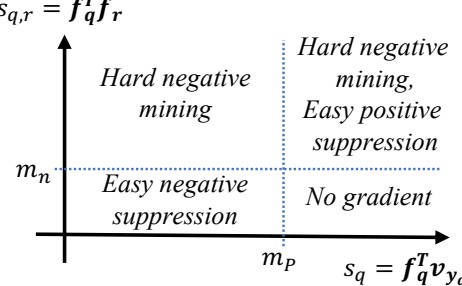

Figure 4: Graph showing regions where hard negative mining, easy positive and negative suppression, and no gradient update occur.

**Property 3.** *Easy negative suppression. If $s_{q,r} - m_n \leq 0$, the derivative of $\mathcal{L}_{D2D\text{-}CE}$ w.r.t $s_{q,r}$ is 0.*

**Property 4.** *If $s_q - m_p \geq 0$ and $s_{q,r} - m_n \leq 0$ are satisfied, $\mathcal{L}_{D2D\text{-}CE}$ has the global minima of $\frac{1}{N} \sum_{i=1}^{N} \log\left(1 + |\mathcal{N}(i)|\right)$.*

We put proofs of the above properties in Appendix D. Property 1 indicates that D2D-CE implicitly conducts hard negative mining and benefits from comparing samples with each other. Also, Properties 2 and 3 imply that samples will not affect gradient updates if the samples are trained sufficiently. Consequently, the classifier concentrates on pushing and pulling hard negative and hard positive examples without being dominated by easy negative and positive samples.

### 3.4 Rebooted Auxiliary Classifier Generative Adversarial Networks (ReACGAN)

With the proposed D2D-CE objective, we propose the *Rebooted Auxiliary Classifier Generative Adversarial Networks* (ReACGAN). As ACGAN does, ReACGAN jointly optimizes an adversarial

loss and the classification objective (D2D-CE). Specifically, the discriminator, which consists of the feature extractor, adversarial head, and linear head, of ReACGAN tries to discriminate whether a given image is sampled from the real distribution or not. At the same time, the discriminator tries to maximize similarities between the reference samples and corresponding proxies while minimizing similarities between negative samples using real images and D2D-CE loss. After updating the discriminator a predetermined number of times, the generator strives to deceive the discriminator by generating well-conditioned images that will output a low D2D-CE value. By alternatively training the discriminator and generator until convergence, ReACGAN generates high-quality images of diverse categories without early-training collapse. We attach the algorithm table in Appendix A.

**Differences between ReACGAN and ContraGAN.** The authors of ContraGAN [23] propose a conditional contrastive loss (2C loss) to cover data-to-data relationships when training cGANs. The main differences between 2C loss and D2D-CE objective are summed up into three points: (1) while 2C loss is derived from NT-Xent loss [39], which is popularly used in the field of the self-supervised learning, our D2D-CE is developed to resolve the early-training collapsing problem and the poor generation results of ACGAN, (2) 2C loss holds false-negative samples in the denominator, and they can cause unexpected positive repulsion forces, and (3) 2C loss contains the similarities between all positive samples in the numerator, and they give rise to large gradients on pulling easy positive samples. Consequently, GANs with 2C loss tend to synthesize images of unintended classes as reported in the author's software document [40]. However, D2D-CE does not contain the false negatives in the denominator and considers only the similarity between a sample and its proxy in the numerator. Therefore, ReACGAN is free from the undesirable repelling forces and does not conduct unnecessary easy positive mining. More detailed explanations are attached in Appendix F.

**Consistency Regularization and D2D-CE Loss.** Zhang *et al.* [16] propose a consistency regularization to force the discriminator to make consistent predictions if given two images are visually close to each other. They create a visually similar image pair by augmenting a reference image with pre-defined augmentations. After that, they let the discriminator minimize L2 distance between the logit of the reference image and the logit of the augmented counterpart. While ReACGAN locates an image embedding nearby its corresponding proxy but far apart multiple image embeddings of different classes, consistency regularization only pulls a reference and the augmented image towards each other. Since consistency regularization and D2D-CE can be applied together, we will show that ReACGAN benefits from consistency regularization in the experiments section (Table 1).

## 4 Experiments

### 4.1 Datasets and Evaluation Metrics

To verify the effectiveness of ReACGAN, we conduct conditional image generation experiments using five datasets: CIFAR10 [30], Tiny-ImageNet [32], CUB200 [33], ImageNet [31], and AFHQ [41] datasets and four evaluation metrics: Inception Score (IS) [42], Fréchet Inception Distance (FID) [34], and $F_{0.125}$ (Precision) and $F_8$ (Recall) [43]. The details on the training datasets are in Appendix C.1.

**Inception Score (IS)** [42] **and Fréchet Inception Distance (FID)** [34] are widely used metrics for evaluating generative models. We utilize IS and FID together because some studies [4, 6, 25] have shown that IS has a tendency to measure the fidelity of images better while FID tends to weight capturing the diversity of images.

**Precision ($F_{0.125}$) and Recall ($F_8$)** [43] are metrics for estimating precision and recall of the approximated distribution $p(G(\mathbf{z}))$ against the true data distribution $p(\mathbf{x})$. Instead of evaluating generative models using one-dimensional scores, such as IS and FID, Sajjadi *et al.* [43] have suggested using a two-dimensional score $F_{0.125}$ and $F_8$ that can quantify how precisely the generated images are and how well the generated images cover the reference distribution.

### 4.2 Experimental Details

The implementation of ReACGAN basically follows details of PyTorch-StudioGAN library [40][1] that supports various experimental setups from ACGAN [18] to StyleGAN2 [7] + ADA [8] with

---

[1]PyTorch-StudioGAN is an open-source library under the MIT license (MIT) with the exception of StyleGAN2 and StyleGAN2 + ADA related implementations, which are under the NVIDIA source code license.

Table 1: Comparison with classifier-based GANs [18, 23] and projection-based GANs [3, 4, 29] on CIFAR10 [30], Tiny-ImageNet [32], and CUB200 [33] datasets using IS [42], FID [34], $F_{0.125}$, and $F_8$ [43] metrics. For baselines, both the numbers from the cited paper (denoted as * in method) and from our experiments using StudioGAN library [40] are reported. The numbers in bold-faced denote the best performance and in underline indicate the values are in one standard deviation from the best.

| Method | CIFAR10 [30] | | | | Tiny-ImageNet [32] | | | | CUB200 [33] | | | |
|---|---|---|---|---|---|---|---|---|---|---|---|---|
| | IS ↑ | FID ↓ | $F_{0.125}$ ↑ | $F_8$ ↑ | IS ↑ | FID ↓ | $F_{0.125}$ ↑ | $F_8$ ↑ | IS ↑ | FID ↓ | $F_{0.125}$ ↑ | $F_8$ ↑ |
| SNGAN* [3] | 8.22 | 21.7 | - | - | - | - | - | - | - | - | - | - |
| BigGAN* [4] | 9.22 | 14.73 | - | - | - | - | - | - | - | - | - | - |
| ContraGAN* [23] | - | 10.60 | - | - | - | 29.49 | - | - | - | - | - | - |
| ACGAN [18] | 9.84 | 8.45 | 0.993 | **0.992** | 6.00 | 96.04 | 0.656 | 0.368 | **6.09** | 60.73 | 0.726 | 0.891 |
| SNGAN [3] | 8.67 | 13.33 | 0.985 | 0.976 | 8.71 | 51.15 | 0.900 | 0.702 | 5.41 | 47.75 | 0.754 | 0.912 |
| SAGAN [29] | 8.66 | 14.31 | 0.983 | 0.973 | 8.74 | 49.90 | 0.872 | 0.712 | 5.48 | 54.29 | 0.728 | 0.882 |
| BigGAN [4] | 9.81 | 8.08 | 0.993 | **0.992** | 12.78 | 32.03 | 0.948 | 0.868 | 4.98 | 18.30 | 0.924 | **0.967** |
| ContraGAN [23] | 9.70 | 8.22 | 0.993 | 0.991 | 13.46 | 28.55 | **0.974** | 0.881 | 5.34 | 21.16 | 0.935 | 0.942 |
| ReACGAN | **9.89** | **7.88** | **0.994** | **0.992** | **14.06** | **27.10** | 0.970 | **0.894** | 4.91 | **15.40** | **0.970** | 0.954 |
| BigGAN + CR* [16] | - | 11.48 | - | - | - | - | - | - | - | - | - | - |
| BigGAN [4] + CR [16] | 9.97 | **7.18** | 0.995 | 0.993 | 15.94 | 19.96 | 0.972 | **0.950** | 5.14 | 11.97 | 0.978 | **0.981** |
| ContraGAN [23] + CR [16] | 9.59 | 8.55 | 0.992 | 0.972 | 15.81 | **19.21** | 0.983 | 0.941 | 4.90 | 11.08 | 0.984 | 0.967 |
| ReACGAN + CR [16] | **10.11** | 7.20 | **0.996** | **0.994** | **16.56** | 19.69 | **0.984** | 0.940 | 4.87 | **10.72** | **0.985** | 0.971 |
| BigGAN + DiffAug* [9] | 9.17 | 8.49 | - | - | - | - | - | - | - | - | - | - |
| BigGAN [4] + DiffAug [9] | 9.94 | 7.17 | 0.995 | 0.992 | 18.08 | 15.70 | 0.980 | **0.972** | 5.53 | 12.15 | 0.967 | 0.981 |
| ContraGAN [23] + DiffAug [9] | 9.95 | 7.27 | 0.995 | 0.992 | 18.20 | 15.40 | 0.986 | 0.963 | 5.39 | 11.02 | 0.978 | 0.970 |
| ReACGAN + DiffAug [9] | **10.22** | **6.79** | **0.996** | **0.993** | **20.60** | **14.25** | **0.988** | **0.972** | 5.22 | **9.27** | **0.985** | **0.983** |
| Real Data | 11.54 | | | | 34.11 | | | | 5.49 | | | |

different scales of datasets [30, 32, 33, 31, 41] and architectures [2, 13, 4, 7]. For a fair comparison, we use the same backbones over all baselines, except otherwise noted, for both the discriminator and generator. For stable training, we apply spectral normalization (SN) [3] to the generator and discriminator except for experiments using SNGAN (in this case, we apply SN to the discriminator only) and StyleGAN2 [7]. We also use the same conditioning method for generators with conditional batch normalization (cBN) [44, 45, 19]. This allows us to investigate solely the conditioning methods of the discriminator, which are the main interest of our paper. If not specified, we use hinge loss [46] as a default for the adversarial loss.

Before conducting main experiments, we perform hyperparameter search with candidates of a temperature $\tau \in \{0.125, 0.25, 0.5, 0.75, 1.0\}$ and a positive margin $m_p \in \{0.5, 0.75, 0.9, 0.95, 0.98, 1.0\}$. We set a negative margin $m_n$ as $1 - m_p$ to reduce search time. Through extensive experiments with 3 runs per each setting, we select $\tau$ with $\{0.5, 0.75, 0.25, 0.5, 0.25\}$ and $m_p$ with $\{0.98, 1.0, 0.95, 0.98, 0.90\}$ for CIFAR10, Tiny-ImageNet, CUB200, ImageNet 256 B.S., and ImageNet 2048 B.S. experiments, respectively. A low temperature seems to work well on fine-grained image generation tasks, but generally, ReACGAN is robust to the choice of hyperparameters. The results of the hyperparameter search and other hyperparameter setups are provided in Appendix C.2.

We evaluate all methods through the same protocol of [9, 16, 17], which uses the same amounts of generated images from the reference split specialized for each dataset.[2] Besides, we run all the experiments three times with random seeds and report the averaged best performances for reliable evaluation with the lone exception of ImageNet and StyleGAN2 related experiments. Please refer to Appendix C.2 for other experimental details. The numbers in bold-faced denote the best performance and in underline indicate that the values are in one standard deviation from the best.

## 4.3 Evaluation Results

**Comparison with Other cGANs.** We compare ReACGAN with previous state-of-the-art cGANs in Tables 1 and 2. We employ the implementations of GANs in PyTorch-StudioGAN library as it provides improved results on standard benchmark datasets [30, 31]. For a fair comparison, we provide results from each original paper (denoted as * in method) as well as those from StudioGAN

---

[2]We use the validation split as the default reference set, but we use the test split of CIFAR10 and the training split of CUB200 and AFHQ due to the absence or lack of the validation dataset.

Table 2: Experiments using ImageNet [31] dataset. B.S. means batch size for training. (Left): Comparisons with previous cGAN approaches. (Right): Learning curves of BigGAN [4], ContraGAN [23], and ReACGAN (ours) which are trained using the batch size of 256.

| | Method | ImageNet [31] | | | |
| | | IS $\uparrow$ | FID $\downarrow$ | $F_{0.125}\uparrow$ | $F_8\uparrow$ |
|---|---|---|---|---|---|
| B.S.=256 | ACGAN* [20] | 7.26 | 184.41 | - | - |
| | SNGAN* [3] | 36.80 | 27.62 | - | - |
| | SAGAN* [29] | 52.52 | 18.28 | - | - |
| | BigGAN* [20] | 38.05 | 22.77 | - | - |
| | TAC-GAN* [20] | 28.86 | 23.75 | - | - |
| | ContraGAN* [23] | 31.10 | 19.69 | 0.951 | 0.927 |
| | ACGAN [18] | 62.99 | 26.35 | 0.935 | 0.963 |
| | SNGAN [3] | 32.25 | 26.79 | 0.938 | 0.913 |
| | SAGAN [29] | 29.85 | 34.73 | 0.849 | 0.914 |
| | BigGAN [4] | 43.97 | 16.36 | 0.964 | 0.955 |
| | ContraGAN [23] | 25.25 | 25.16 | 0.947 | 0.855 |
| | ReACGAN | **68.27** | **13.98** | **0.976** | **0.977** |
| | BigGAN [4] + DiffAug [9] | 36.97 | 18.57 | 0.956 | 0.941 |
| | ReACGAN + DiffAug [9] | **69.74** | **11.95** | **0.977** | **0.975** |
| B.S.=2048 | BigGAN* [4] | 99.31 | 8.51 | - | - |
| | BigGAN* [25] | **104.57** | 9.18 | - | - |
| | BigGAN [4] | 99.71 | **7.89** | 0.985 | 0.989 |
| | ReACGAN | 92.74 | 8.23 | **0.991** | **0.990** |
| | Real Data | 173.33 | | | |

IS and FID scores during training

library [40]. We also conduct experiments with popular augmentation-based methods: consistency regularization (CR) [16] and differentiable augmentation (DiffAug) [9].

Compared with other cGANs, ReACGAN performs the best on most benchmarks, surpassing the previous methods by 2.5%, 5.1%, 15.8%, and 14.5% in FID on CIFAR10, Tiny-ImageNet, CUB200, and ImageNet (256 B.S.), respectively. ReACGAN also harmonizes with augmentation-based regularizations, bringing incremental improvements on all the metrics. For the ImageNet experiments using a batch size of 256, ReACGAN reaches higher IS and lower FID with fewer iterations than other models in comparison. Finally, we demonstrate that ReACGAN can learn with a larger batch size on ImageNet. While some recent methods [20, 38, 27] have been built on ACGAN to improve the generation performance of ACGAN, large-scale image generation experiments with the batch size of 2048 have never been reported. Table 2 shows that our ReACGAN reaches FID score of 8.23 on ImageNet, being comparable with the value of 7.89 from BigGAN implementation in PyTorch-StudioGAN library. ReACGAN, however, provides better synthesis results than other implementations of BigGAN [4, 25]. Note that our result on ImageNet is obtained in only two runs while the training setup and architecture of BigGAN have been extensively searched and finely tuned.

**Comparison with Other Conditioning Losses.** We investigate how the generation qualities vary with different conditioning losses while keeping the other configurations fixed. We compare D2D-CE loss with cross-entropy loss of ACGAN (AC) [18], loss used in the projection discriminator (PD) [19], multi-hinge loss (MH) [26], and conditional contrastive loss (2C) [23]. The results are shown in Table 3. AC- and MH-based models present decent results on CIFAR10, but undergo early-training collapse on Tiny-ImageNet and CUB200 datasets. Replacing them with PD, 2C, and D2D-CE losses produce satisfactory performances across all datasets, where PD loss makes the best $F_8$ (recall) on CUB200 dataset while giving third $F_{0.125}$ (precision) value. The noticeable point is that the proposed D2D-CE shows consistent results across all datasets, showing the lowest FID and the highest $F_8$ and $F_{0.125}$ values in most cases. This means ReACGAN can generate high-fidelity images and is relatively free from the precision and recall trade-off [43] than the others.

**Consistent Performance of ReACGAN on Adversarial Loss Selection.** We validate the consistent performance of ReACGAN on four adversarial losses: non-saturation loss [1], least square loss [10], Wasserstein loss with gradient penalty regularization (W-GP) [13], and hinge loss [46] on CIFAR10 and Tiny-ImageNet datasets in Table 4. The experimental results show that BigGAN + D2D-CE (ReACGAN) consistently outperforms the projection discriminator (PD) and conditional contrastive loss (2C) counterparts over three adversarial losses [1, 13, 46]. However, for the experiments using the least square loss [10], ReACGAN exhibits inferior generation performances to the projection discriminator. We speculate that minimizing the least square distance between an adversarial logit and the target scalar (1 or 0) might affect the norms of feature maps and spoil the classifier training performed by D2D-CE loss.

Table 3: Experiments on the effectiveness of D2D-CE loss compared with other conditioning losses.

| Conditioning Method | CIFAR10 [30] | | | | Tiny-ImageNet [32] | | | | CUB200 [33] | | | |
|---|---|---|---|---|---|---|---|---|---|---|---|---|
| | IS ↑ | FID ↓ | $F_{0.125}$ ↑ | $F_8$ ↑ | IS ↑ | FID ↓ | $F_{0.125}$ ↑ | $F_8$ ↑ | IS ↑ | FID ↓ | $F_{0.125}$ ↑ | $F_8$ ↑ |
| BigGAN w/o Condition [4] (Abbreviated to Big) | 9.46 | 12.21 | 0.987 | 0.982 | 7.38 | 76.15 | 0.804 | 0.576 | 5.16 | 35.17 | 0.852 | 0.936 |
| Big + AC [18] | 9.84 | 8.45 | 0.993 | **0.992** | 6.00 | 96.04 | 0.656 | 0.368 | **6.09** | 60.73 | 0.726 | 0.891 |
| Big + PD [19] | 9.81 | 8.08 | 0.993 | **0.992** | 12.78 | 32.03 | 0.948 | 0.868 | 4.98 | 18.30 | 0.924 | **0.967** |
| Big + MH [26] | **10.05** | 7.94 | **0.994** | 0.990 | 4.37 | 140.74 | 0.282 | 0.156 | 5.18 | 245.69 | 0.625 | 0.832 |
| Big + 2C [23] | 9.70 | 8.22 | 0.993 | 0.991 | 13.46 | 28.55 | **0.974** | 0.881 | 5.34 | 21.16 | 0.935 | 0.942 |
| Big + D2D-CE (ReACGAN) | 9.89 | **7.88** | **0.994** | **0.992** | **14.06** | **27.10** | 0.970 | **0.894** | 4.91 | **15.40** | **0.970** | 0.954 |

Table 4: Experiments to identify the consistent performance of D2D-CE on adversarial loss selection.

| Adversarial Loss | Conditioning Method | CIFAR10 [30] | | | | | Tiny-ImageNet [32] | | | | |
|---|---|---|---|---|---|---|---|---|---|---|---|
| | | IS ↑ | FID ↓ | $F_{0.125}$ ↑ | $F_8$ ↑ | Better? | IS ↑ | FID ↓ | $F_{0.125}$ ↑ | $F_8$ ↑ | Better? |
| Non-saturation [1] | PD [19] | 9.75 | 8.29 | **0.993** | 0.991 | ✓ | 8.27 | 58.85 | 0.816 | 0.713 | |
| | 2C [23] | 9.30 | 10.47 | 0.990 | 0.959 | | 6.57 | 84.27 | 0.745 | 0.556 | |
| | D2D-CE | **9.79** | **8.27** | **0.993** | 0.991 | ✓ | **11.76** | **39.32** | **0.942** | **0.852** | ✓ |
| Least square [10] | PD [19] | **9.94** | **8.26** | **0.993** | **0.992** | ✓ | **12.74** | **37.14** | **0.920** | **0.900** | ✓ |
| | 2C [23] | 8.66 | 12.18 | 0.986 | 0.941 | | 9.58 | 53.10 | **0.916** | 0.706 | |
| | D2D-CE | 9.70 | 9.56 | 0.991 | 0.987 | | 9.50 | 57.67 | 0.848 | 0.692 | |
| W-GP [13] | PD [19] | 5.71 | 64.75 | 0.792 | 0.652 | | 6.67 | 84.16 | 0.696 | 0.498 | |
| | 2C [23] | 5.93 | 55.99 | 0.842 | 0.709 | | 6.89 | 74.45 | 0.812 | 0.536 | |
| | D2D-CE | **7.30** | **35.94** | **0.942** | **0.847** | ✓ | **8.92** | **52.74** | **0.856** | **0.689** | ✓ |
| Hinge [46] | PD [19] | 9.81 | 8.08 | 0.993 | **0.992** | | 12.78 | 32.03 | 0.948 | 0.868 | |
| | 2C [23] | 9.70 | 8.22 | 0.993 | 0.991 | | 13.46 | 28.55 | **0.974** | 0.881 | |
| | D2D-CE | **9.89** | **7.88** | **0.994** | **0.992** | ✓ | **14.06** | **27.10** | 0.970 | **0.894** | ✓ |

Table 5: FID [34] values of conditional StyleGAN2 (cStyleGAN2) [7] and StyleGAN2 [7] + D2D-CE on CIFAR10 [30] and AFHQ [41] datasets. ADA means the adaptive discriminator augmentation [8]. FID is exceptionally computed using the training split for calculating the reference moments since FID value of StyleGAN2 is often calculated using the moments of the training dataset.

| Conditioning method | CIFAR10 [30] | AFHQ [41] |
|---|---|---|
| cStyleGAN2 [7] | 3.88 | - |
| StyleGAN2 [7] + D2D-CE | **3.34** | - |
| cStyleGAN2 [7] + ADA [8] | 2.43 | 4.99 |
| StyleGAN2 [7] + ADA [8] + D2D-CE | **2.38** | **4.95** |
| StyleGAN2 [7] + DiffAug [9] + D2D-CE + Tuning | **2.26** | - |

**Effect of D2D-CE for Different GAN Architectures.** We study the effect of D2D-CE loss with different GAN architectures. In Table 5, we validate that D2D-CE is effective for StyleGAN2 [7] backbone and also fits well with the adaptive discriminator augmentation (ADA) [8]. StyleGAN2 with D2D-CE loss produces 13.9% better generation result than the conditional version of StyleGAN2 (cStyleGAN2) on CIFAR10. Moreover, StyleGAN2 with D2D-CE can be reinforced with ADA or DiffAug when train StyleGAN2 + D2D-CE under the limited data situation. Among GANs, StyleGAN2 + DiffAug + D2D-CE + Tuning achieves the best performance on CIFAR10, even outperforming some diffusion-based methods [47, 48]. Additional results with other architectures, i.e., a deep convolutional network [2] and a resnet style backbone [13], are provided in Appendix E.

## 4.4 Ablation Study

We study how each component of ReACGAN affects ACGAN training. By adding or ablating each part of ReACGAN, as shown in Table 6, we identify four major observations. (1) Feature and weight normalization greatly stabilize ACGAN training and improve generation performances on Tiny-ImageNet and CUB200 datasets. (2) D2D-CE enhances the generation performance by considering data-to-data relationships and by performing easy sample suppression (3th and 4th rows). (3) the suppression technique does not work well on the feature normalized cross-entropy loss (5th row). (4) While ACGAN shows high Inception score on ImageNet experiment, it shows relatively

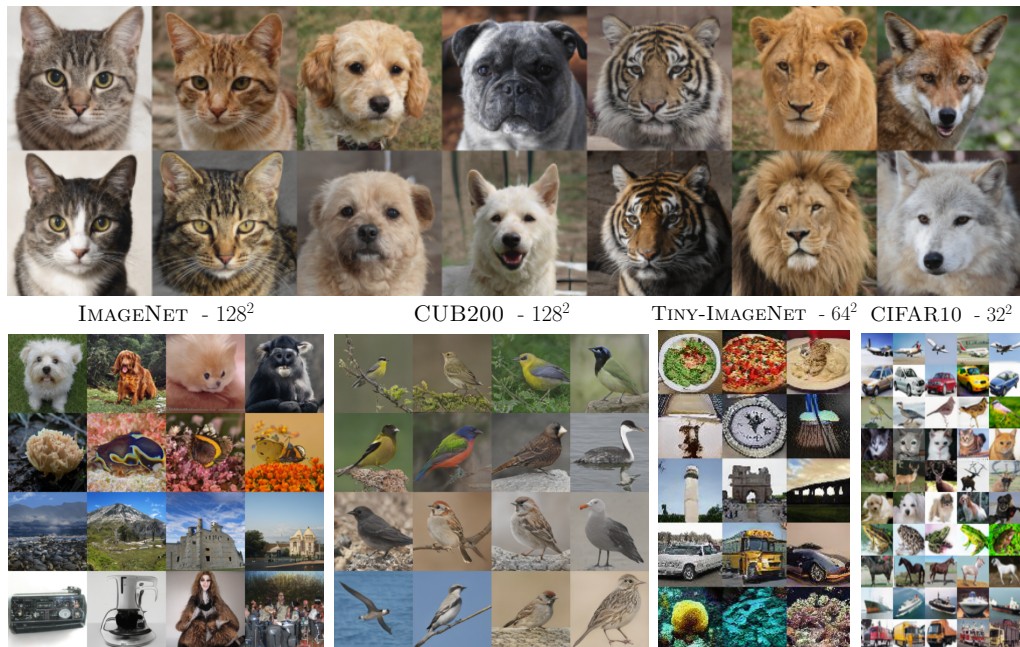

Figure 5: Curated images generated by the proposed ReACGAN. More qualitative results on ReAC-GAN, BigGAN [4], ContraGAN [23], and ACGAN [18] are attached in Appendix J.

Table 6: Ablation study on feature map normalization, data-to-data consideration, and easy positive and negative sample suppression.

| Ablation | Tiny-ImageNet [32] | | | | CUB200 [33] | | | | ImageNet [49] | | | |
|---|---|---|---|---|---|---|---|---|---|---|---|---|
| | IS ↑ | FID ↓ | $F_{0.125}$ ↑ | $F_8$ ↑ | IS ↑ | FID ↓ | $F_{0.125}$ ↑ | $F_8$ ↑ | IS ↑ | FID ↓ | $F_{0.125}$ ↑ | $F_8$ ↑ |
| ACGAN [18] | 6.00 | 96.04 | 0.656 | 0.368 | **6.09** | 60.73 | 0.726 | 0.891 | 62.99 | 26.35 | 0.935 | 0.963 |
| + Normalization | 13.46 | 30.33 | 0.955 | 0.889 | 4.78 | 25.54 | 0.883 | 0.952 | 18.16 | 36.40 | 0.879 | 0.787 |
| + Data-to-data (Eq. (5)) | 12.96 | 28.71 | 0.967 | 0.863 | 5.08 | 25.12 | 0.894 | 0.946 | - | - | - | - |
| + Suppression (Eq. (6)) | **14.06** | **27.10** | **0.970** | **0.894** | 4.91 | **15.40** | **0.970** | **0.954** | **63.16** | **14.59** | **0.974** | **0.974** |
| - Data-to-data | 12.96 | 30.79 | 0.960 | 0.857 | 5.39 | 30.36 | 0.863 | 0.947 | - | - | - | - |

poor FID, $F_{0.125}$, and $F_8$ values compared with the model trained with normalization, data-to-data consideration, and the suppression technique. This result is consistent with the qualitative results on ImageNet, where the images from ACGAN are easily classifiable, but the images from ReACGAN are high quality and diverse. We attribute the improvement to the discriminator that successfully leverages informative data-to-data and data-to-class relationships with easy sample suppression.

## 5 Conclusion

In this paper, we have analyzed why training ACGAN becomes unstable as the number of classes in the dataset increases. By deriving the analytic form of gradient in the classifier and numerically checking the gradient values, we have discovered that the unstable training comes from a gradient exploding problem caused by the unboundedness of input feature vectors and poor classification ability of the classifier in the early training stage. To alleviate the instability and reinforce ACGAN, we have proposed the Data-to-Data Cross-Entropy loss (D2D-CE) and the Rebooted Auxiliary Classifier Generative Adversarial Network (ReACGAN). The experimental results verify the superiority of ReACGAN compared with the existing classifier- and projection-based GANs on five benchmark datasets. Moreover, exhaustive analyses on ReACGAN prove that ReACGAN is robust to hyperparameter selection and harmonizes with various architectures and differentiable augmentations.

## Acknowledgments and Disclosure of Funding

This work was supported by the IITP grants (No.2019-0-01906: AI Graduate School Program - POSTECH, No.2021-0-00537: Visual Common Sense, No.2021-0-02068: AI Innovation Hub) funded by Ministry of Science and ICT, Korea.

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
