# Appendices:
# Rebooting ACGAN: Auxiliary Classifier GANs with Stable Training

## A  Algorithm

---

**Algorithm 1** : Training ReACGAN

---

**Input:** Batch size: $N$. Temperature: $\tau$. Margin values: $m_p, m_n$. Balance coefficient: $\lambda$.
Parameters of the generator: $\theta$.
Parameters of the discriminator (feature extractor + adversarial head + linear head): $\phi$.
\# of discriminator updates per single generator update: $n_{dis}$.
Adversarial loss: $\mathcal{L}_{\text{Adv}}$ [1, 2, 3, 4].
Learning rate: $\alpha_1, \alpha_2$. Adam hyperparameters [5]: $\beta_1, \beta_2$.
**Output:** Optimized $(\theta, \phi)$.

1: Initialize $(\theta, \phi)$
2: **for** $\{1, ..., \text{\# of training iterations}\}$ **do**
3:    **for** $\{1, ..., n_{\text{dis}}\}$ **do**
4:       Sample $\boldsymbol{X}, \boldsymbol{y}^{\text{real}} = \{\boldsymbol{x}_i\}_{i=1}^N, \{y_i\}_{i=1}^N \sim p_{\text{real}}(\mathbf{x}, \mathbf{y})$
5:       Sample $\boldsymbol{Z} = \{\boldsymbol{z}_i\}_{i=1}^N \sim p(\mathbf{z})$ and $\boldsymbol{y}^{\text{fake}} = \{y_i^{\text{fake}}\}_{i=1}^N \sim P(\mathbf{y})$
6:       $\mathcal{L}_{\text{D\_Adv}} \longleftarrow \mathcal{L}_{\text{Adv}}(\boldsymbol{X}, \boldsymbol{y}^{\text{real}}, G(\boldsymbol{Z}, \boldsymbol{y}^{\text{fake}}), \boldsymbol{y}^{\text{fake}})$
7:       $\mathcal{L}_{\text{D\_Cond}} \longleftarrow \mathcal{L}_{\text{D2D-CE}}(\boldsymbol{X}, \boldsymbol{y}^{\text{real}}; \tau, m_p, m_n)$            ▷ Eq. (6) with real images.
8:       $\phi \longleftarrow \text{Adam}(\mathcal{L}_{\text{D\_Adv}} + \lambda \mathcal{L}_{\text{D\_Cond}}, \alpha_1, \beta_1, \beta_2)$
9:    **end for**
10:   Sample $\boldsymbol{Z} = \{\boldsymbol{z}_i\}_{i=1}^N \sim p(\mathbf{z})$ and $\boldsymbol{y}^{\text{fake}} = \{y_i^{\text{fake}}\}_{i=1}^N \sim P(\mathbf{y})$
11:   $\mathcal{L}_{\text{G\_Adv}} \longleftarrow \mathcal{L}_{\text{Adv}}(G(\boldsymbol{Z}, \boldsymbol{y}^{\text{fake}}), \boldsymbol{y}^{\text{fake}})$
12:   $\mathcal{L}_{\text{G\_Cond}} \longleftarrow \mathcal{L}_{\text{D2D-CE}}(G(\boldsymbol{Z}, \boldsymbol{y}^{\text{fake}}), \boldsymbol{y}^{\text{fake}}; \tau, m_p, m_n)$            ▷ Eq. (6) with fake images.
13:   $\theta \longleftarrow \text{Adam}(\mathcal{L}_{\text{G\_Adv}} + \lambda \mathcal{L}_{\text{G\_Cond}}, \alpha_2, \beta_1, \beta_2)$
14: **end for**

---

## B  Software: PyTorch-StudioGAN

Generative Adversarial Network (GAN) is one of the popular generative models for realistic image generation. Although GAN has been actively studied in the machine learning community, only a few open-source libraries provide reliable implementations for GAN training. In addition, the existing libraries do not support various training and test configurations for loss functions, backbone architectures, regularizations, differentiable augmentations, and evaluation metrics. In this paper, we expand StudioGAN [6] library, and the StudioGAN provides about 40 implementations of GAN-related papers as follows:

**GANs**: DCGAN [7], LSGAN [2], GGAN [4], WGAN-WC [8], WGAN-GP [3], WGAN-DRA [9], ACGAN [10], Projection discriminator [11], SNGAN [12], SAGAN [13], TACGAN [14], LGAN [15], BigGAN [16], BigGAN-deep [16], StyleGAN2 [17], CRGAN [18], ICRGAN [19], LOGAN [20], ContraGAN [21], MHGAN [22], ADCGAN [23], ReACGAN (ours).

**Adversarial losses**: Logistic loss [17], Non-saturation loss [1], Least square loss [2], Wasserstein loss [8], Hinge loss [4], Multiple discriminator loss [24], Multi-hinge loss [22].

**Regularizations**: Feature matching regularization [25], R1 regularization [24], Weight clipping regularization [8], Spectral normalization [12], Path length regularization [26, 17], Top-k training [27].

**Metrics**: IS [25], FID [28], Intra-class FID, CAS [29], Precision and recall [30], Improved precision and recall [31], Density and coverage [32], SwAV backbone FID [33].

**Differentiable augmentations**: SimCLR augmentation [34, 35], BYOL augmentation [36, 35], DiffAugment [37], Adaptive discriminator augmentation (ADA) [38].

**Miscellaneous**: Mixed precision training [39], Distributed data parallel (DDP), Data parallel (DP), Synchronized batch normalization [40], Standing statistics [16], Truncation trick [16, 17], Freeze discriminator (FreezeD) [41], Discriminator driven latent sampling (DDLS) [42], Closed-form factorization (SeFa) [43].

## C   Training Details

### C.1. Datasets

**CIFAR10** [44] is a widely used benchmark dataset for evaluating cGANs. The dataset contains 60k $32 \times 32$ RGB images which belong to 10 different classes. The dataset is split into 50k images for training and 10k images for testing.

**Tiny-ImageNet** [45] contains 120k $64 \times 64$ RGB images and is split into 100k training, 10k validation, and 10k test images. Tiny-ImageNet consists of 200 categories, and training GANs on Tiny-ImageNet is more challenging than CIFAR10 since there is less data (500 images) per class.

**CUB200** [46] provides around 12k fine-grained RGB images for 200 bird classes. We apply the center crop to each image using a square box whose lengths are the same as the short side of the image, and we resize the images to $128 \times 128$ pixels. We train cGANs on CUB200 dataset to identify the generation ability of cGANs on images with fine-grained characteristics in a limited data situation.

**ImageNet** [47] provides around 1,281k and 50k RGB images for training and validation. We preprocess each image in the same way as applied to CUB200.

**AFHQ** [48] consists of 14,630 and 1,500 numbers of $512 \times 512$ RGB images for training and validation. The dataset is divided into 3 different animal classes (cat, dog, and wild animals).

For training and testing, we apply horizontal flip augmentation for all datasets and normalize image pixel values to a range between -1 and 1.

### C.2. Hyperparameter Setup

Selecting proper hyperparameter values greatly affects GAN training. So it might be helpful to specify details of hyperparameter setups used in our work for future study. In this section, we aim to provide training specifications as much as possible, and if there exists a missing experimental setup, it follows configurations and details of StudioGAN implementation [6].

Table A1 shows hyperparameter setups used in our experiments. The settings (A, C, E, G) are used for baseline experiments on CIFAR10: BigGAN [16], BigGAN with DiffAug [37], StyleGAN2 [17], and StyleGAN2 with ADA [38], the setting (I) on Tiny-ImageNet: BigGAN and BigGAN with DiffAug, the setting (K) on CUB200: BigGAN and BigGAN with DiffAug, the settings (M, N, P) on ImageNet: ACGAN/SNGAN/ContraGAN, BigGAN/ReACGAN/DiffAug-BigGAN/DiffAug-ReACGAN with a batch size of 256, and BigGAN with a batch size of 2048, and the setting (R) on AFHQ: StyleGAN2 with ADA. For ReACGAN experiments, we utilize the settings (B, D, F, H, J, L, O, Q, S) for experiments on the datasets stated above. To select an appropriate temperature $\tau$ and positive margin $m_p$, we conduct two-stage linear search with the candidates of a temperature $\tau \in \{0.125, 0.25, 0.5, 0.75, 1.0\}$ and a positive margin $m_p \in \{0.5, 0.75, 0.9, 0.95, 0.98, 1.0\}$ while fixing the dimensionalities of feature embeddings to 512, 768, 1024, and 2048 for CIFAR10, Tiny-ImageNet, CUB200, and ImageNet experiments. We set the balance coefficient $\lambda$ equal to the temperature $\tau$ except for ImageNet generation experiments. Specifically, we explore the best temperature value on each dataset and fix the temperature for the linear search on the positive margin. Linear search results are summarized in Fig. A1, and the results show that ReACGAN provides stable performances across various temperature and positive margin values.

Table A1: Hyperparameter setups for cGAN training. The settings (A, C, E, I, K, M, P, R) are commonly used practices in previous studies [13, 16, 21, 38]

| Setting | Batch size | Adam($\alpha_1, \alpha_2, \beta_1, \beta_2$) [5] | $n_{dis}$ | $(\lambda, \tau)$ | $m_p$ | G_Ema [49] | Ema start | Total iterations |
|---|---|---|---|---|---|---|---|---|
| A | 64 | (2e-4, 2e-4, 0.5, 0.999) | 5 | - | - | True | 1k | 100k |
| B | 128 | (2.82e-4, 2.82e-4, 0.5, 0.999) | 5 | (0.5, 0.5) | 0.98 | True | 1k | 100k |
| C | 64 | (2e-4, 2e-4, 0.5, 0.999) | 5 | - | - | True | 1k | 200k |
| D | 128 | (2.82e-4, 2.82e-4, 0.5, 0.999) | 5 | (0.5, 0.5) | 0.98 | True | 1k | 200k |
| E | 64 | (2.5e-3, 2.5e-3, 0.0, 0.99) | 1 | - | - | True | 0 | 200k |
| F | 64 | (2.5e-3, 2.5e-3, 0.0, 0.99) | 2 | (0.25, 0.25) | 0.98 | True | 0 | 200k |
| G | 64 | (2.5e-3, 2.5e-3, 0.0, 0.99) | 1 | - | - | True | 0 | 800k |
| H | 64 | (2.5e-3, 2.5e-3, 0.0, 0.99) | 2 | (0.25, 0.25) | 0.98 | True | 0 | 800k |
| I | 1024 | (4e-4, 1e-4, 0.0, 0.999) | 1 | - | - | True | 20k | 100k |
| J | 1024 | (4e-4, 1e-4, 0.0, 0.999) | 1 | (0.75, 0.75) | 1.0 | True | 20k | 100k |
| K | 256 | (2e-4, 5e-5, 0.0, 0.999) | 2 | - | - | True | 4k | 40k |
| L | 256 | (2e-4, 5e-5, 0.0, 0.999) | 2 | (0.25, 0.25) | 0.95 | True | 4k | 40k |
| M | 256 | (2e-4, 5e-5, 0.0, 0.999) | 2 | - | - | True | 20k | 200k |
| N | 256 | (2e-4, 5e-5, 0.0, 0.999) | 2 | - | - | True | 20k | 600k |
| O | 256 | (2e-4, 5e-5, 0.0, 0.999) | 2 | (1.0, 0.5) | 0.98 | True | 20k | 600k |
| P | 2048 | (2e-4, 5e-5, 0.0, 0.999) | 2 | - | - | True | 20k | 200k |
| Q | 2048 | (2e-4, 5e-5, 0.0, 0.999) | 2 | (0.5, 0.25) | 0.90 | True | 20k | 200k |
| R | 64 | (2.5e-3, 2.5e-3, 0.0, 0.99) | 1 | - | - | True | 0 | 200k |
| S | 64 | (2.5e-3, 2.5e-3, 0.0, 0.99) | 2 | (0.5, 0.5) | 0.95 | True | 0 | 200k |

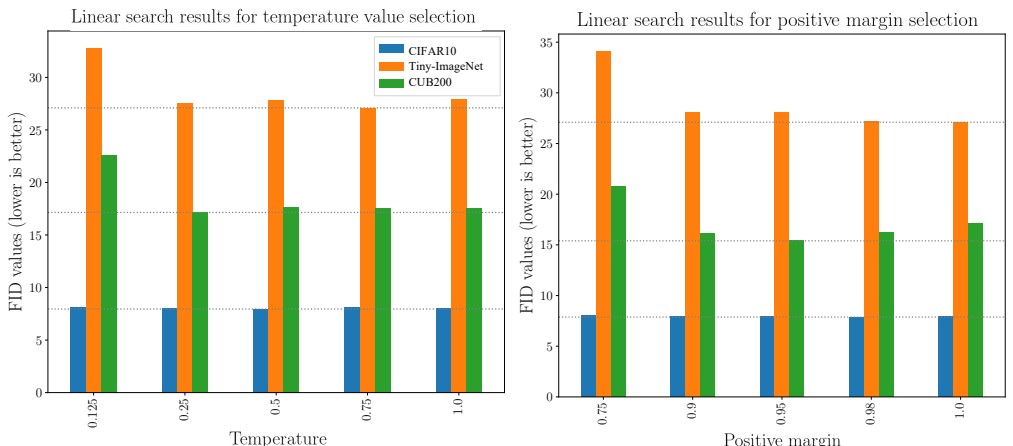

Figure A1: FID [28] values of ReACGANs with different temperatures and positive margins. The dotted lines indicate the best performances on each dataset.

## D   Proofs of Properties of D2D-CE

In the main paper, we introduce a new objective, the D2D-CE to train ACGAN stably. In this section, we provide proofs of the properties of D2D-CE. Using the notations defined in the main paper, our proposed D2D-CE loss can be expressed as follows:

$$\mathcal{L}_{\text{D2D-CE}} = -\frac{1}{N} \sum_{i=1}^{N} \log \left( \frac{\exp\left([\boldsymbol{f}_i^\top \boldsymbol{v}_{y_i} - m_p]_- / \tau\right)}{\exp\left([\boldsymbol{f}_i^\top \boldsymbol{v}_{y_i} - m_p]_- / \tau\right) + \sum_{j \in \mathcal{N}(i)} \exp\left([\boldsymbol{f}_i^\top \boldsymbol{f}_j - m_n]_+ / \tau\right)} \right), \text{ (A1)}$$

where $i$ is a sample index and $\mathcal{N}(i)$ is the set of indices that indicate the locations of negative samples in the mini-batch. To understand properties of D2D-CE loss, we can rewrite Eq. (A1) as follows:

$$\mathcal{L}_{\text{D2D-CE}} = -\frac{1}{N} \sum_{i=1}^{N} \log \left( \frac{\exp\left([s_i - m_p]_- / \tau\right)}{\exp\left([s_i - m_p]_- / \tau\right) + \sum_{j \in \mathcal{N}(i)} \exp\left([s_{i,j} - m_n]_+ / \tau\right)} \right). \quad \text{(A2)}$$

Let $s_q$ be a similarity between a normalized reference sample embedding $\boldsymbol{f}_q$ and the corresponding normalized proxy $\boldsymbol{v}_{y_q}$, $s_{q,r}$ be a similarity between $\boldsymbol{f}_q$ and one of the its negative samples $\boldsymbol{f}_r$, and $a, b \in \mathcal{N}(q)$ be arbitrary indices of negative samples. Then, we can summarize the four properties of D2D-CE loss as follows:

**Property 1.** *Hard negative mining. If the value of $s_{q,a}$ is greater than $s_{q,b}$, the derivative of $\mathcal{L}_{D2D\text{-}CE}$ w.r.t $s_{q,a}$ is greater than or equal to the derivative w.r.t $s_{q,b}$; that is $\frac{\partial \mathcal{L}_{D2D\text{-}CE}}{\partial s_{q,a}} \geq \frac{\partial \mathcal{L}_{D2D\text{-}CE}}{\partial s_{q,b}} \geq 0$.*

**Property 2.** *Positive suppression. If $s_q - m_p \geq 0$, the derivative of $\mathcal{L}_{D2D\text{-}CE}$ w.r.t $s_q$ is 0.*

**Property 3.** *Negative suppression. If $s_{q,r} - m_n \leq 0$, the derivative of $\mathcal{L}_{D2D\text{-}CE}$ w.r.t $s_{q,r}$ is 0.*

**Property 4.** *If $s_q - m_p \geq 0$ and $s_{q,r} - m_n \leq 0$ are satisfied, $\mathcal{L}_{D2D\text{-}CE}$ has the global minima of $\frac{1}{N}\sum_{i=1}^{N}\log\left(1 + |\mathcal{N}(i)|\right)$.*

***Proof of Property 1.*** By expanding Eq. (A2), we have the following equation:

$$\mathcal{L}_{\text{D2D-CE}} = -\underbrace{\frac{1}{\tau N}\sum_{i=1}^{N}[s_i - m_p]_-}_{\text{Positive attraction}}$$
$$+ \underbrace{\frac{1}{N}\sum_{i=1}^{N}\log\left(\exp\left([s_i - m_p]_-/\tau\right) + \sum_{j\in\mathcal{N}(i)}\exp\left([s_{i,j} - m_n]_+/\tau\right)\right)}_{\text{Negative repulsion}}. \tag{A3}$$

Based on this, we calculate the derivative of $\mathcal{L}_{\text{D2D-CE}}$ w.r.t $s_{q,r}$ as follows:

$$\frac{\partial \mathcal{L}_{\text{D2D-CE}}}{\partial s_{q,r}} = \frac{\mathbf{1}_{s_{i,j}-m_n>0}(i=q, j=r)\exp\left((s_{q,r}-m_n)/\tau\right)}{\tau N\left(\exp\left([s_q-m_p]_-/\tau\right) + \sum_{j\in\mathcal{N}(q)}\exp\left([s_{q,j}-m_n]_+/\tau\right)\right)}. \tag{A4}$$

Since we assume $s_{q,a} > s_{q,b}$ is satisfied, the derivative of $\frac{\partial \mathcal{L}_{\text{D2D-CE}}}{\partial s_{q,a}}$ is greater than $\frac{\partial \mathcal{L}_{\text{D2D-CE}}}{\partial s_{q,b}}$ except when the indicator functions $\mathbf{1}_{s_{i,j}-m_n>0}(i=q, j=a)$ and $\mathbf{1}_{s_{i,j}-m_n>0}(i=q, j=b)$ are 0. Note that the value of the derivative $\frac{\partial \mathcal{L}_{\text{D2D-CE}}}{\partial s_{q,r}}$ exponentially increases as the similarity $s_{q,r}$ linearly increases, and this means that $\mathcal{L}_{\text{D2D-CE}}$ conducts hard negative mining.

***Proof of Property 2.*** Based on Eq. (A3), we can derive the derivative of $\mathcal{L}_{\text{D2D-CE}}$ w.r.t $s_q$ as follows:

$$\frac{\partial \mathcal{L}_{\text{D2D-CE}}}{\partial s_q} = -\frac{1}{\tau N}\mathbf{1}_{s_i-m_p<0}(i=q)$$
$$+ \frac{\mathbf{1}_{s_i-m_p<0}(i=q)\exp\left((s_q-m_p)/\tau\right)}{\tau N\left(\exp\left([s_q-m_p]_-/\tau\right) + \sum_{j\in\mathcal{N}(q)}\exp\left([s_{q,j}-m_n]_+/\tau\right)\right)}. \tag{A5}$$

Since the indicator function $\mathbf{1}_{s_i-m_p<0}(i=q)$ gives 0 value when $s_q - m_p \geq 0$, the derivative of $\mathcal{L}_{\text{D2D-CE}}$ w.r.t $s_q$ is 0 when $s_q - m_p \geq 0$ is satisfied.

***Proof of Property 3.*** Based on Eq. (A4), the derivative of $\mathcal{L}_{\text{D2D-CE}}$ w.r.t $s_{q,r}$ is 0 when $s_{q,r} - m_n \leq 0$.

***Proof of Property 4.*** We can get the global minima of $\mathcal{L}_{\text{D2D-CE}}$ by plugging in 0 values inside of the exponential components in Eq. (A3) as follows:

$$\mathcal{L}_{\text{D2D-CE}} = \frac{1}{N}\sum_{i=1}^{N}\log\left(\exp\left(0\right) + \sum_{j\in\mathcal{N}(i)}\exp\left(0\right)\right)$$
$$= \frac{1}{N}\sum_{i=1}^{N}\log\left(1 + |\mathcal{N}(i)|\right). \tag{A6}$$

# E  Additional Experimental Results

**Gradients Exploding Problem in ACGAN.** We conduct additional experiments regarding the gradient exploding problem of ACGAN using CUB200 dataset, and the results can be seen in Fig. A2.

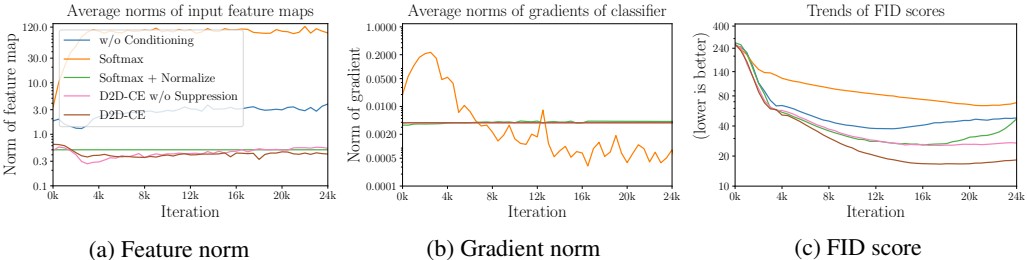

| (a) Feature norm | (b) Gradient norm | (c) FID score |

Figure A2: Merits of integrating feature normalization and data-to-data relationship consideration. The experiments are conducted using CUB200 [46] dataset. (a) Average norms of input feature maps, (b) average norms of gradients of classification losses, and (c) trends of FID scores. Compared to AC-GAN [10], the proposed ReACGAN does not experience the early-training collapse problem caused by excessively large norms of feature maps and gradients. In addition, ReACGAN can converge to a better equilibrium by considering data-to-data relationships with easy sample suppression.

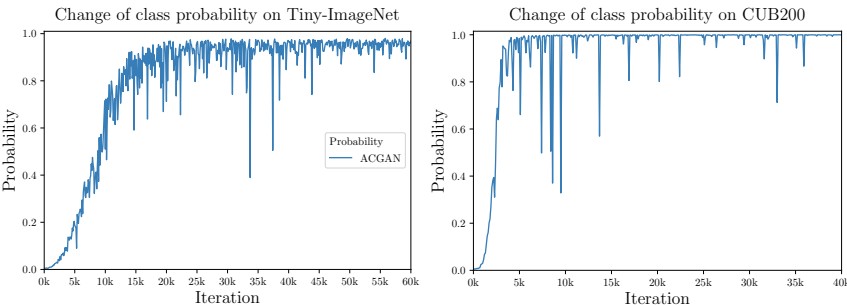

Figure A3: Trends of the probability values from the classifiers trained on Tiny-ImageNet [45] and CUB200 [46] datasets.

Same as the experimental results using Tiny-ImageNet (Fig. 2 in the main paper), normalizing feature maps resolves the early-training collapse problem. Also, considering data-to-data relationships brings extra performance gain with easy positive and negative sample suppression.

**ACGAN Focuses on Classifier Training instead of Adversarial Learning.** To identify ACGAN is prone to being biased toward classifying categories of images instead of discriminating the authenticity of given samples, we track the trend of classifier's target probabilities as the training progresses using Tiny-ImageNet and CUB200 datasets. As can be seen in Fig. A2, A3, and Fig. 2 in the main paper, classifier's target probabilities continuously increase, but the FID scores do not decrease as of certain points in time. Therefore the experimental results imply that ACGAN training is likely to become biased toward label classification instead of adversarial training.

**Can ReACGAN Approximate a Mixture of Gaussian Distributions Whose Supports Overlap?** We conduct distribution approximation experiments using a 1-D mixture of Gaussian distributions (MoG). The experiments are proposed by Gong *et al.* [14] and are devised to identify if a given GAN can estimate any true data distribution, even the mixture of Gaussians with overlapped supports. Although ReACGAN has shown successful outputs on real images, ReACGAN fails to estimate the 1-D MoG, resulting in poor approximation ability similar to ACGAN (see Fig. A4). However, this phenomenon is not weird because ReACGAN follows the same optimization process as ACGAN does, which inherently reduces a conditional entropy $\mathcal{H}(\mathbf{y}|\mathbf{x})$. As the result, ACGAN can only accurately approximate marginal distributions generated by conditional distributions with non-overlapped supports.

To deal with this problem, Gong *et al.* [14] have suggested using a twin auxiliary classifier (TAC) on the top of the discriminator and have demonstrated that TAC enables ACGAN to exactly estimate the 1-D MoG. Therefore, ReACGAN can be reinforced by introducing TAC, and the experimental result verifies that ReACGAN can approximate the 1-D MoG exactly (see Fig. A4).

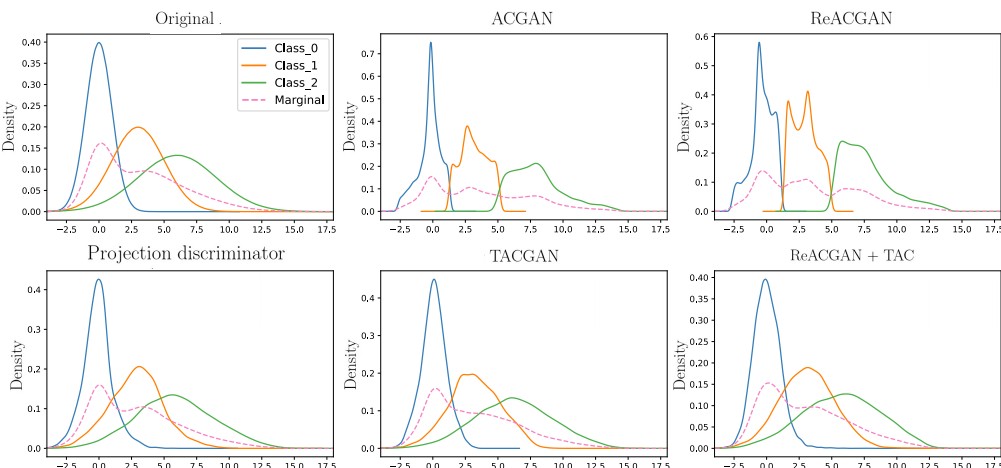

Figure A4: Comparison with ACGAN [10], Projection discriminator [11], TACGAN [14], ReAC-GAN, and ReACGAN + TAC on a synthetic 1-D MoG dataset [14]. We conduct all experiments using the same setup specified in the paper [14].

Table A2: Experiments to identify the effectiveness of ReACGAN with TAC [14] on CIFAR10 [44] and Tiny-ImageNet [45] datasets.

| Dataset | ACGAN [10] | | TACGAN [14] | | ReACGAN | | ReACGAN + TAC [14] | |
| --- | --- | --- | --- | --- | --- | --- | --- | --- |
| | IS ↑ | FID ↓ | IS ↑ | FID ↓ | IS ↑ | FID ↓ | IS ↑ | FID ↓ |
| CIFAR10 [44] | 9.84 | 8.45 | 9.78 | 8.01 | **9.89** | **7.88** | 9.70 | 7.94 |
| Tiny-ImageNet [45] | 6.00 | 96.04 | 7.62 | 65.99 | **14.06** | 27.10 | 13.71 | **26.14** |

Finally, to validate the effectiveness of TAC for ReACGAN on real datasets, we perform CIFAR10 and Tiny-ImageNet generation experiments. Contrary to our expectations, Table A2 shows that ReACGAN + TAC provides comparable or marginally better results on CIFAR10 and Tiny-ImageNet datasets over the ReACGAN. We speculate that this is because benchmark datasets are highly refined and might follow a mixture of non-overlapped conditional distributions.

**Effect of D2D-CE for Different GAN Architectures.** We perform additional experiments to identify the effect of D2D-CE loss for different GAN architectures. We utilize a deep convolutional neural network (Deep CNN) [7] and a ResNet-style network (ResNet) [3] to train GANs on CIFAR10 and Tiny-IamgeNet datasets. The experimental results show that ReACGAN provides consistent generation results on different architectures (see Table A3).

Table A3: Experiments for investigating the effect of D2D-CE for different architectures using CIFAR10 [44] and Tiny-ImageNet [45] datasets. We report only FID [28] for a compact expression.

| Conditioning method | Deep CNN [7] on CIFAR10 [44] | ResNet [3] on CIFAR10 [44] | ResNet [3] on Tiny-ImageNet [45] |
| --- | --- | --- | --- |
| AC [10] | 20.35 | 13.04 | 87.84 |
| PD [11] | 19.49 | 13.47 | 47.88 |
| 2C [21] | 21.47 | 14.38 | **40.56** |
| D2D-CE (ReACGAN) | **18.94** | **12.47** | 40.89 |

**Effect of Number of Negative Samples on ReACGAN Training.** We investigate how the number of negative samples affects the generation performance of ReACGAN using CIFAR10 and Tiny-ImageNet datasets. First, we compute pairwise similarities between all negative samples in the mini-batch. Then, we drop similarities between negative samples using a randomly generated mask whose element has a value of 0 according to the pre-defined probability $p$ and otherwise has a value of 1. For example, $p = 0.1$ will lead approximately 10% of the total similarities between negative samples not to account for calculating the denominator part of D2D-CE loss. As can be

Table A4: Ablation study on the number of negative samples. FID score [28] is used for evaluation.

| Dataset | Masking probability $p$ for negative samples in Eq. (6) | | | | | |
|---|---|---|---|---|---|---|
| | $p = 1.0$ | 0.8 | 0.6 | 0.4 | 0.2 | 0.0 |
| CIFAR10 [44] | 11.15 | 8.07 | 8.11 | 7.83 | 8.01 | 7.88 |
| Tiny-ImageNet [45] | 60.74 | 30.04 | 28.64 | 28.59 | 28.68 | 27.10 |

seen in Table A4, D2D-CE loss benefits from more negative samples. This implies that the more the data-to-data relations are provided, the richer supervision signals for conditioning become, resulting in better image generation results. In addition, from the optimization point of view, the generator and discriminator can receive gradient signals at an exponential rate as the number of negative samples increases linearly.

**Are There Any Possible Prescriptions for Preventing the Gradient Exploding Problem?** In the main paper, we verify that simply normalizing feature embeddings onto a unit hypersphere resolves ACGAN's early-training collapse problem. In this section, we explore if there exist other cures for resolving the early-training collapse problem: (1) lowering classification strength, (2) gradient clipping, and (3) feature clipping. The experimental results (Table A5) indicate that lowering classification strength and normalizing feature maps can prevent ACGAN training from collapsing at the early training phase. However, we cannot succeed in training ACGAN by clipping gradients of the classifier. We speculate that this is because gradient clipping restricts not only the norms of feature maps but also the class probability values; thus, ACGAN can be updated by inaccurate gradients and ends up collapsing. Among those methods, the normalization and D2D-CE present better performances than the others, demonstrating the effectiveness of our proposals.

Table A5: Experiments for studying available cures for preventing the gradient exploding problem in ACGAN. FID [28] scores are reported for evaluation. $\lambda$ is a balance coefficient between adversarial learning and classifier training.

| Dataset | $\lambda = 0.25$ | $\lambda = 0.5$ | $\lambda = 0.75$ | $\lambda = 1$ | Normalization | Feature clipping | Gradient clipping | D2D-CE |
|---|---|---|---|---|---|---|---|---|
| CIFAR100 [44] | 12.30 | 13.61 | 15.60 | 16.92 | 13.17 | 17.47 | 40.23 | **12.25** |
| Tiny-ImageNet [45] | 62.86 | 92.05 | 104.34 | 98.75 | 28.04 | 57.65 | 108.30 | **27.10** |

**Training Time per 100 Generator Updates.** We investigate training times of BigGAN, ContraGAN, and ReACGAN on ImageNet using 8 Nvidia V100 GPUs. The batch size is set to 2048. We identify that ReACGAN brings in a slight computational overhead and takes about 1.05∼1.1 longer time than the other GANs. Specifically, BigGAN takes 17m 37s, ContraGAN 18m 24s, and ReACGAN 18m 52s per 100 generator updates.

# F   Analysis of the differences between ReACGAN and ContraGAN

This section explains the differences between ReACGAN and ContraGAN [21] from a mathematical point of view. Kang and Park [21] have proposed the conditional contrastive loss (2C loss), which is formulated from NT-Xent loss [34], and developed contrastive generative adversarial networks (ContraGAN) for conditional image generation. Using the notations used in our main paper, we can write down 2C loss as follows:

$$\mathcal{L}_{2C} = -\frac{1}{N} \sum_{i=1}^{N} \log \left( \frac{\exp\left(\boldsymbol{f}_i^\top \boldsymbol{v}_{y_i}/\tau\right) + \sum_{n_p \in \mathcal{P}(i)} \exp(\boldsymbol{f}_i^\top \boldsymbol{f}_{n_p}/\tau)}{\exp\left(\boldsymbol{f}_i^\top \boldsymbol{v}_{y_i}/\tau\right) + \sum_{j \in \{1,...,N\}\setminus\{i\}} \exp\left(\boldsymbol{f}_i^\top \boldsymbol{f}_j/\tau\right)} \right), \qquad (A7)$$

where $\mathcal{P}(i)$ is the set of indices that indicate locations of positive samples in the mini-batch. To clearly identify how each sample embedding updates, we start by considering 2C loss on a single

sample. We can rewrite a single sample version of Eq. (A7) as follows:

$$\mathcal{L}'_{2C} = -\underbrace{\frac{1}{N}\log\left(\exp\left(\boldsymbol{f}_q^\top \boldsymbol{v}_{y_q}/\tau\right) + \sum_{n_p\in\mathcal{P}(q)}\exp(\boldsymbol{f}_q^\top \boldsymbol{f}_{n_p}/\tau)\right)}_{\text{Positive attraction}}$$

$$+\underbrace{\frac{1}{N}\log\left(\exp\left(\boldsymbol{f}_q^\top \boldsymbol{v}_{y_q}/\tau\right) + \sum_{j\in\{1,...,N\}\setminus\{q\}}\exp\left(\boldsymbol{f}_q^\top \boldsymbol{f}_j/\tau\right)\right)}_{\text{Negative repulsion}}. \tag{A8}$$

Using Eq. (A8), we can calculate the derivative of $\mathcal{L}'_{2C}$ w.r.t $\boldsymbol{f}_q$ as follows:

$$\frac{\partial\mathcal{L}'_{2C}}{\partial\boldsymbol{f}_q} = -\frac{\exp(\boldsymbol{f}_q^\top \boldsymbol{v}_{y_q}/\tau)\boldsymbol{v}_{y_q} + \sum_{n_p\in\mathcal{P}(q)}\exp\left(\boldsymbol{f}_q^\top \boldsymbol{f}_{n_p}/\tau\right)\boldsymbol{f}_{n_p}}{\tau N\left(\exp(\boldsymbol{f}_q^\top \boldsymbol{v}_{y_q}/\tau) + \sum_{n_p\in\mathcal{P}(q)}\exp\left(\boldsymbol{f}_q^\top \boldsymbol{f}_{n_p}/\tau\right)\right)}$$

$$+\frac{\exp(\boldsymbol{f}_q^\top \boldsymbol{v}_{y_q}/\tau)\boldsymbol{v}_{y_q} + \sum_{j\in\{1,...,N\}\setminus\{q\}}\exp\left(\boldsymbol{f}_q^\top \boldsymbol{f}_j/\tau\right)\boldsymbol{f}_j}{\tau N\left(\exp(\boldsymbol{f}_q^\top \boldsymbol{v}_{y_q}/\tau) + \sum_{j\in\{1,...,N\}\setminus\{q\}}\exp\left(\boldsymbol{f}_q^\top \boldsymbol{f}_j/\tau\right)\right)}. \tag{A9}$$

To understand Eq. (A9) more intuitively, we replace the denominator terms of Eq. (A9) with $A$ and $B$ and re-organize the equation as follows:

$$\frac{\partial\mathcal{L}_{2C}}{\partial\boldsymbol{f}_q} = -\frac{\exp(\boldsymbol{f}_q^\top \boldsymbol{v}_{y_q}/\tau)\boldsymbol{v}_{y_q} + \sum_{n_p\in\mathcal{P}(q)}\exp\left(\boldsymbol{f}_q^\top \boldsymbol{f}_{n_p}/\tau\right)\boldsymbol{f}_{n_p}}{\tau NA}$$

$$+\frac{\exp(\boldsymbol{f}_q^\top \boldsymbol{v}_{y_q}/\tau)\boldsymbol{v}_{y_q} + \sum_{j\in\{1,...,N\}\setminus\{q\}}\exp\left(\boldsymbol{f}_q^\top \boldsymbol{f}_j/\tau\right)\boldsymbol{f}_j}{\tau NB}$$

$$= -\underbrace{\frac{1}{\tau N}\left(\frac{\exp(\boldsymbol{f}_q^\top \boldsymbol{v}_{y_q}/\tau)\boldsymbol{v}_{y_q} + \overbrace{\sum_{n_p\in\mathcal{P}(q)}\exp\left(\boldsymbol{f}_q^\top \boldsymbol{f}_{n_p}/\tau\right)\boldsymbol{f}_{n_p}}^{\text{(P) Positive samples}}}{A} - \frac{\exp(\boldsymbol{f}_q^\top \boldsymbol{v}_{y_q}/\tau)\boldsymbol{v}_{y_q}}{B}\right)}_{\text{Positive attraction}}$$

$$+\underbrace{\frac{1}{\tau N}\left(\frac{\sum_{n_q\in\mathcal{N}(q)}\exp\left(\boldsymbol{f}_q^\top \boldsymbol{f}_{n_q}/\tau\right)\boldsymbol{f}_{n_q} + \overbrace{\sum_{j\in\{1,...,N\}\setminus(\{q\}\cup\mathcal{N}(q))}\exp\left(\boldsymbol{f}_q^\top \boldsymbol{f}_j/\tau\right)\boldsymbol{f}_j}^{\text{(F) False negative samples}}}{B}\right)}_{\text{Negative repulsion}}. \tag{A10}$$

The above equation implies that the positive samples (P) in Eq (A10) can cause easy positive mining, *i.e.*, if a similarity $\boldsymbol{f}_q^\top \boldsymbol{f}_{n_p}$ has a large value, the gradient $\frac{\partial\mathcal{L}_{2C}}{\partial\boldsymbol{f}_q}$ can be biased towards $\boldsymbol{f}_{n_p}$ direction with large magnitude. In addition, the false-negative samples (F) can attenuate the negative repulsion force, which is already being addressed in the contrastive learning community [50, 51, 52]. Unlike 2C loss, our D2D-CE loss does not experience the easy positive mining and the attenuation caused by false-negative samples. To demonstrate this, we write down D2D-CE loss as follows:

$$\mathcal{L}_{\text{D2D-CE}} = -\frac{1}{N}\sum_{i=1}^{N}\log\left(\frac{\exp\left([\boldsymbol{f}_i^\top \boldsymbol{v}_{y_i} - m_p]_-/\tau\right)}{\exp\left([\boldsymbol{f}_i^\top \boldsymbol{v}_{y_i} - m_p]_-/\tau\right) + \sum_{j\in\mathcal{N}(i)}\exp\left([\boldsymbol{f}_i^\top \boldsymbol{f}_j - m_n]_+/\tau\right)}\right). \tag{A11}$$

In the same way as before, we can expand a single sample version of Eq. (A11) as follows:

$$\mathcal{L}_{\text{D2D-CE}} = - \underbrace{\frac{1}{\tau N} [\boldsymbol{f}_q^\top \boldsymbol{v}_{y_q} - m_p]_-}_{\text{Positive attraction}}$$

$$+ \underbrace{\frac{1}{N} \log \left( \exp \left( [\boldsymbol{f}_q^\top \boldsymbol{v}_{y_q} - m_p]_- / \tau \right) + \sum_{j \in \mathcal{N}(q)} \exp \left( [\boldsymbol{f}_q^\top \boldsymbol{f}_j - m_n]_+ / \tau \right) \right)}_{\text{Negative repulsion}}. \quad \text{(A12)}$$

Based on Eq. (A12), we can calculate the derivative of $\mathcal{L}_{\text{D2D-CE}}$ w.r.t $\boldsymbol{f}_q$ as follows:

$$\frac{\partial \mathcal{L}_{\text{D2D-CE}}}{\partial \boldsymbol{f}_q} = - \frac{1}{\tau N} \mathbf{1}_{\boldsymbol{f}_i^\top \boldsymbol{v}_{y_i} - m_p < 0}(i = q) \boldsymbol{v}_{y_q}$$

$$+ \frac{1}{\tau N} \left( \frac{\mathbf{1}_{\boldsymbol{f}_i^\top \boldsymbol{v}_{y_i} - m_p < 0}(i = q) \exp \left( (\boldsymbol{f}_q^\top \boldsymbol{v}_{y_q} - m_p)/\tau \right) \boldsymbol{v}_{y_q}}{\exp \left( [\boldsymbol{f}_q^\top \boldsymbol{v}_{y_q} - m_p]_- / \tau \right) + \sum_{j \in \mathcal{N}(q)} \exp \left( [\boldsymbol{f}_q^\top \boldsymbol{f}_j - m_n]_+ / \tau \right)} \right)$$

$$+ \frac{1}{\tau N} \left( \frac{\sum_{j \in \mathcal{N}(q)} \mathbf{1}_{\boldsymbol{f}_i^\top \boldsymbol{f}_j - m_n > 0}(i = q) \exp \left( (\boldsymbol{f}_q^\top \boldsymbol{f}_j - m_n)/\tau \right) \boldsymbol{f}_j}{\exp \left( [\boldsymbol{f}_q^\top \boldsymbol{v}_{y_q} - m_p]_- / \tau \right) + \sum_{j \in \mathcal{N}(q)} \exp \left( [\boldsymbol{f}_q^\top \boldsymbol{f}_j - m_n]_+ / \tau \right)} \right)$$

$$= - \underbrace{\frac{\mathbf{1}_{\boldsymbol{f}_i^\top \boldsymbol{v}_{y_i} - m_p < 0}(i = q)}{\tau N} \left( \boldsymbol{v}_{y_q} - \frac{\exp \left( (\boldsymbol{f}_q^\top \boldsymbol{v}_{y_q} - m_p)/\tau \right) \boldsymbol{v}_{y_q}}{C} \right)}_{\text{Positive attraction}}$$

$$+ \underbrace{\frac{1}{\tau N} \left( \frac{\sum_{j \in \mathcal{N}(q)} \mathbf{1}_{\boldsymbol{f}_i^\top \boldsymbol{f}_j - m_n > 0}(i = q) \exp \left( (\boldsymbol{f}_q^\top \boldsymbol{f}_j - m_n)/\tau \right) \boldsymbol{f}_j}{C} \right)}_{\text{Negative repulsion}}, \quad \text{(A13)}$$

where $C = \exp \left( [\boldsymbol{f}_q^\top \boldsymbol{v}_{y_q} - m_p]_- / \tau \right) + \sum_{j \in \mathcal{N}(q)} \exp \left( [\boldsymbol{f}_q^\top \boldsymbol{f}_j - m_n]_+ / \tau \right)$. Unlike 2C loss, D2D-CE loss does not contain multiple positive samples in the positive attraction bracket and only consists of negative samples in the negative repulsion part, which means that D2D-CE loss does not perform easy-positive mining and does not attenuate the negative repulsion force.

Table A6: Top-1 and Top-5 ImageNet classification accuracies on generated images from AC-GAN [10], BigGAN [16], ContraGAN [21], and ReACGAN (ours). We use ImageNet pre-trained Inception-V3 model [53] as a classifier. To generate images from GANs, we use the best checkpoints saved during 200k generator updates.

| | Real data (validation) | ACGAN [10] | BigGAN [16] | ContraGAN [21] | ReACGAN |
|---|---|---|---|---|---|
| Top-1 Accuracy (%) | 70.822 | **62.412** | 29.994 | 2.866 | 23.210 |
| Top-5 Accuracy (%) | 89.574 | **84.899** | **53.842** | 11.482 | 51.602 |
| IS [25] ↑ | 173.33 | **62.99** | 28.63 | 25.25 | 50.30 |
| FID [28] ↓ | - | 26.35 | 24.68 | 25.16 | **16.32** |

To compare the conditioning performance of ReACGAN with other cGANs, we calculate Top-1 and Top-5 classification accuracies on ImageNet [47] using the pre-trained Inception-V3 network [53]. The results are summarized in Table A6. Although ReACGAN has a lower FID value and higher IS score compared with BigGAN [16] and ContraGAN [21], the top-1 and top-5 accuracies of ReACGAN are slightly below that of BigGAN. This implies that ReACGAN tends to approximate overall distribution with a slight loss of the exact conditioning. On the other hand, ContraGAN fails to perform conditional image generation, and it provides 2.866 % Top-1 accuracy on ImageNet dataset. This indicates that ContraGAN is likely to generate undesirably conditioned but visually satisfactory images (see Fig. A10, A14, A18, and A21 for quantitative results). One more interesting point is that although generated images from ACGAN give the best classification accuracy, they show a poor FID value compared with the others. This implies that ACGAN generates well-classifiable images without considering the diversity and fidelity of generated samples (see Fig. A12).

# G   Potential Negative Societal Impacts

The success in generating photo-realistic images in GANs [16, 26, 17] has attracted a myriad of applications to be developed, such as photo editing (filtering [54], stylization [55] and object removal [56]), image translation (sketch → clip art [57], photo → cartoon [58]), image in-painting [59], and image extrapolation to arbitrary resolutions [60]. While, in most cases, GANs are helpful for content creation or fast prototyping, there exist potential threats that one can maliciously use the synthesized results to deceive others. A well-known example is deepfake [61], where a person in the video appears with the voice and appearance of a celebrity and conveys a message to deceive or confuse others, *e.g.*, fake news. Other examples include sexual harnesses [62] and hacking machine vision applications.

As an effort to circumvent the negative issues, a number of techniques have been proposed. Masi *et al.* [63] have utilized color and frequency information to detect deepfake. Naseer *et al.* [64] have developed a general defense method from self-attacking via feature perturbation. We anticipate that further development of synthetic image detection techniques, well-established policies on the technique, and ethical awareness of researchers/developers will enable us to enjoy the broad applicability and benefits of GANs.

# H   Computation resources

In this section, we provide a summary of the total number of performed experiments, computing resources, and approximated training time spent on our research in Table A7. Since we have conducted a lot of experiments with various configurations using different resources, we divide our experiments into 16 divisions and calculate *approximate time spent* on each division of experiments.

Table A7: Approximate total training time (days) provided for reference.

| Division of experiments | GPU Type | Days | # of experiments | Approximate Time (days) |
|---|---|---|---|---|
| CIFAR10 [44] | RTX 2080 Ti | 0.75 | 100 | 75 |
| CIFAR10 [44] + CR [18] | RTX 2080 Ti | 1.17 | 9 | 10.53 |
| CIFAR10 [44] + DiffAug [37] | RTX 2080 Ti | 2.04 | 9 | 18.36 |
| CIFAR10 [44] + StyleGAN2 [17] | TITAN Xp×2 | 2.58 | 2 | 5.16 |
| CIFAR10 [44] + StyleGAN2 [17] + ADA [38] | TITAN Xp×2 | 9.52 | 2 | 19.04 |
| Tiny-ImageNet [45] | TITAN RTX×4 | 1.54 | 84 | 132.44 |
| Tiny-ImageNet [45] + CR [18] | TITAN RTX×4 | 1.42 | 9 | 12.78 |
| Tiny-ImageNet [45] + DiffAug [37] | TITAN RTX×4 | 2.83 | 9 | 25.47 |
| CUB200 [46] | TITAN RTX×4 | 1.63 | 24 | 39.12 |
| CUB200 [46] + CR [18] | TITAN RTX×4 | 0.92 | 9 | 8.28 |
| CUB200 [46] + DiffAug [37] | TITAN RTX×4 | 0.67 | 9 | 6.03 |
| ImageNet [47] (200k iter., B.S.=256) | Tesla V100×4 | 4.17 | 6 | 25.02 |
| ImageNet [47] (600k iter., B.S.=256) | Tesla V100×4 | 12.51 | 2 | 25.02 |
| ImageNet [47] + DiffAug [37] (600k iter., B.S.=256) | A100×4 | 12.43 | 2 | 24.86 |
| ImageNet [47] (B.S.=2048) | Tesla V100×8 | 26.90 | 2 | 53.80 |
| AFHQ [48] + StyleGAN2 [17] + ADA [38] | A100×4 | 1.96 | 2 | 3.92 |
| Total | | | 280 | 484.83 |

# I   Standard Deviations of Experiments

We run all the experiments three times with random seeds and report the averaged best performances for reliable evaluation with the lone exception of ImageNet experiments. This section provides standard deviations for reference.

Table A8: Comparisons with classifier-based GANs [10, 21] and projection-based GANs [12, 16, 13] on CIFAR10 [44], Tiny-ImageNet [45], and CUB200 [46] datasets using IS [25], FID [28], $F_{0.125}$ and $F_8$ [30] metrics. We report the standard deviations of three different runs in this table.

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

Table A11: Ablation study on normalization, data-to-data consideration, and easy negative and positive sample suppression.

| Ablation | Tiny-ImageNet [45] | | | | CUB200 [46] | | | |
|---|---|---|---|---|---|---|---|---|
| | IS ↑ | FID ↓ | $F_{0.125}$ ↑ | $F_8$ ↑ | IS ↑ | FID ↓ | $F_{0.125}$ ↑ | $F_8$ ↑ |
| ACGAN [10] | 0.12 | 8.04 | 0.021 | 0.048 | 0.73 | 9.20 | 0.036 | 0.102 |
| + Normalization | - | - | - | - | - | - | - | - |
| + Data-to-data (Eq. (5)) | - | - | - | - | - | - | - | - |
| + Suppression (Eq. (6)) | 0.07 | 0.56 | 0.003 | 0.009 | 0.09 | 0.80 | 0.007 | 0.009 |
| - Data-to-data | - | - | - | - | - | - | - | - |

Table A12: Experiments for investigating the effect of D2D-CE for different architectures using CIFAR10 [44] and Tiny-ImageNet [45] datasets.

| Conditioning method | Deep CNN [7] on CIFAR10 [44] | ResNet [3] on CIFAR10 [44] | ResNet [3] on Tiny-ImageNet [45] |
|---|---|---|---|
| AC [10] | 0.02 | 0.11 | 0.34 |
| PD [11] | 0.67 | 0.25 | 1.29 |
| 2C [21] | 1.28 | 1.33 | 0.18 |
| D2D-CE (ReACGAN) | 0.03 | 0.23 | 0.29 |

Table A13: Ablation study on the number of negative samples.

| Dataset | Masking probability $p$ for negative samples in Eq. (6) | | | | | |
|---|---|---|---|---|---|---|
| | $p = 1.0$ | 0.8 | 0.6 | 0.4 | 0.2 | 0.0 |
| CIFAR10 [44] | 0.33 | 0.04 | 0.11 | 0.13 | 0.05 | 0.07 |
| Tiny-ImageNet [45] | 8.62 | 1.82 | 0.76 | 0.92 | 0.87 | 0.56 |

# J  Qualitative Results

We provide images that are generated by our ReACGAN and baseline approaches (ContraGAN [21], BigGAN [16], and ACGAN [10]).

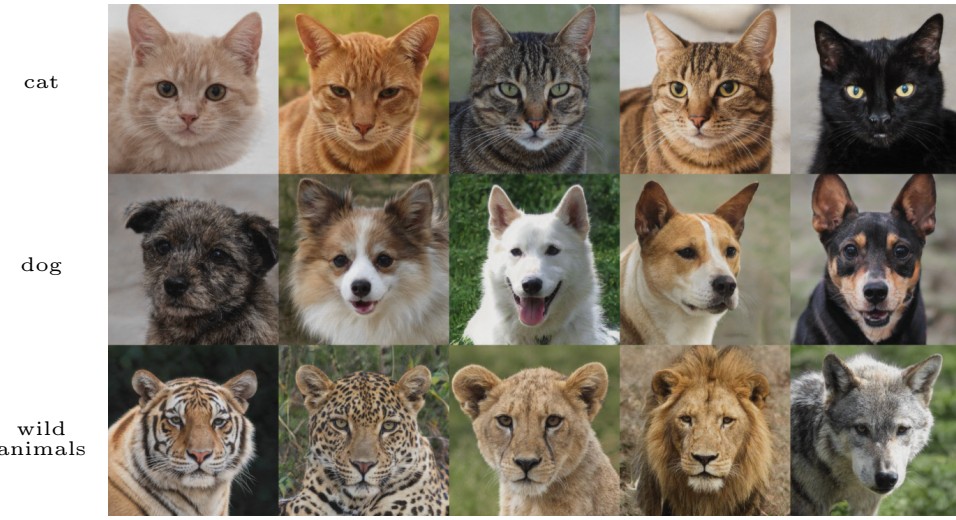

Figure A5: Generated images on AFHQ [48] dataset using StyleGAN2 [17] + ADA [38] + D2D-CE (ReACGAN) (FID=4.95).

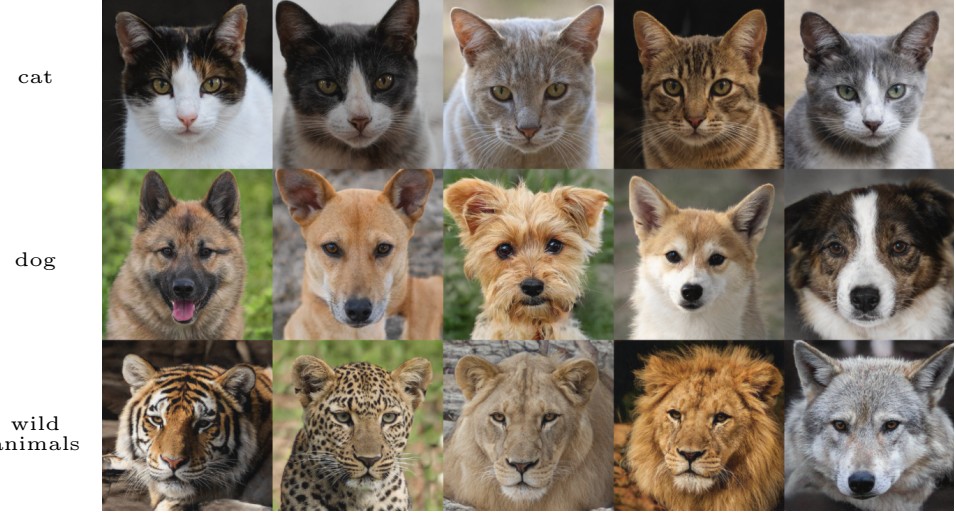

Figure A6: Generated images on AFHQ [48] dataset using StyleGAN2 [17] + ADA [38] (FID=4.99).

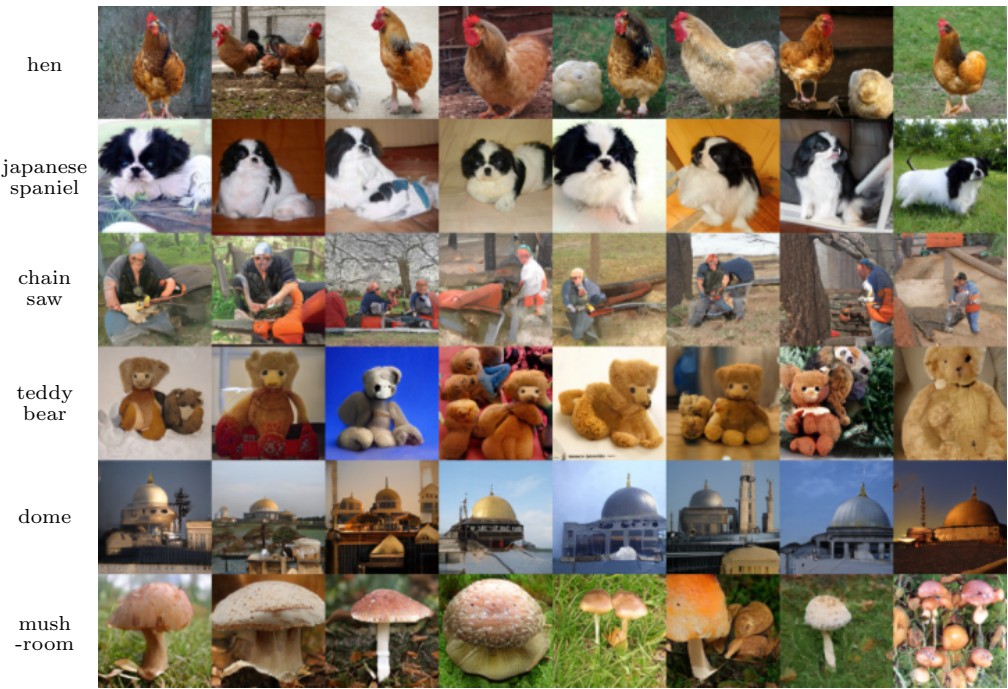

Figure A7: Generated images on ImageNet [47] dataset using ReACGAN and the batch size of 2048 (FID=8.23).

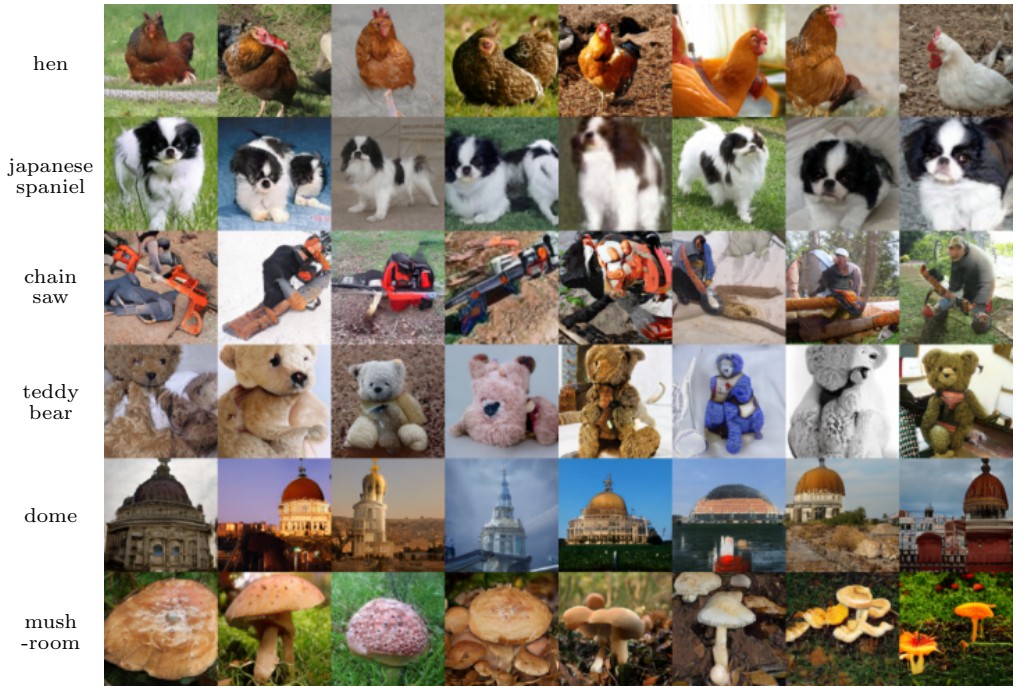

Figure A8: Generated images on ImageNet [47] dataset using BigGAN [16] and the batch size of 2048 (FID=7.89).

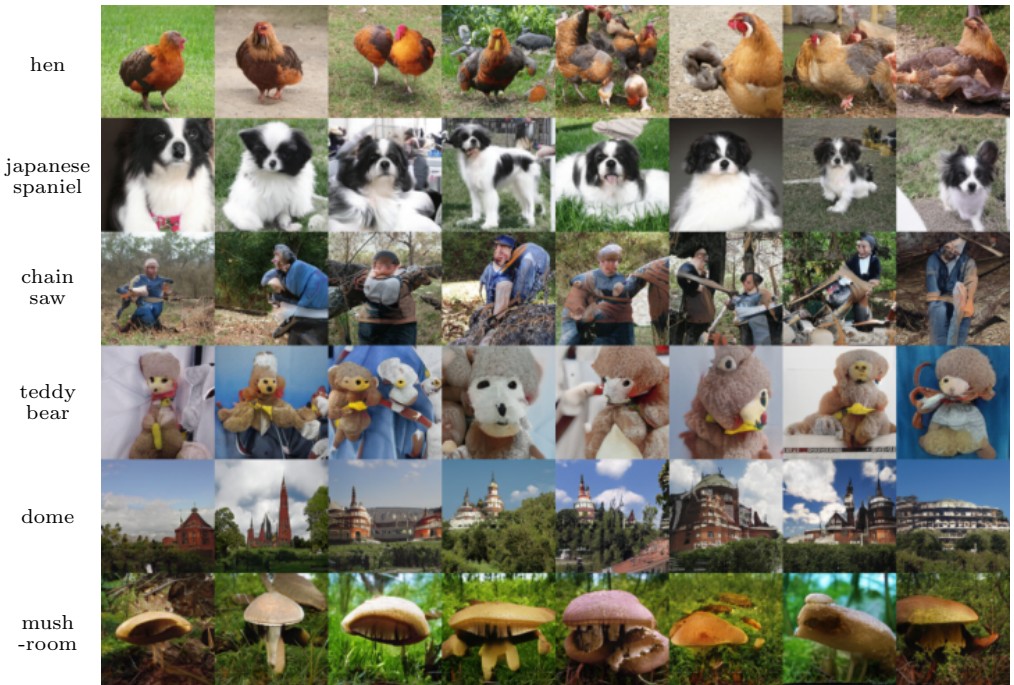

Figure A9: Generated images on ImageNet [47] dataset using ReACGAN and the batch size of 256 (FID=13.98).

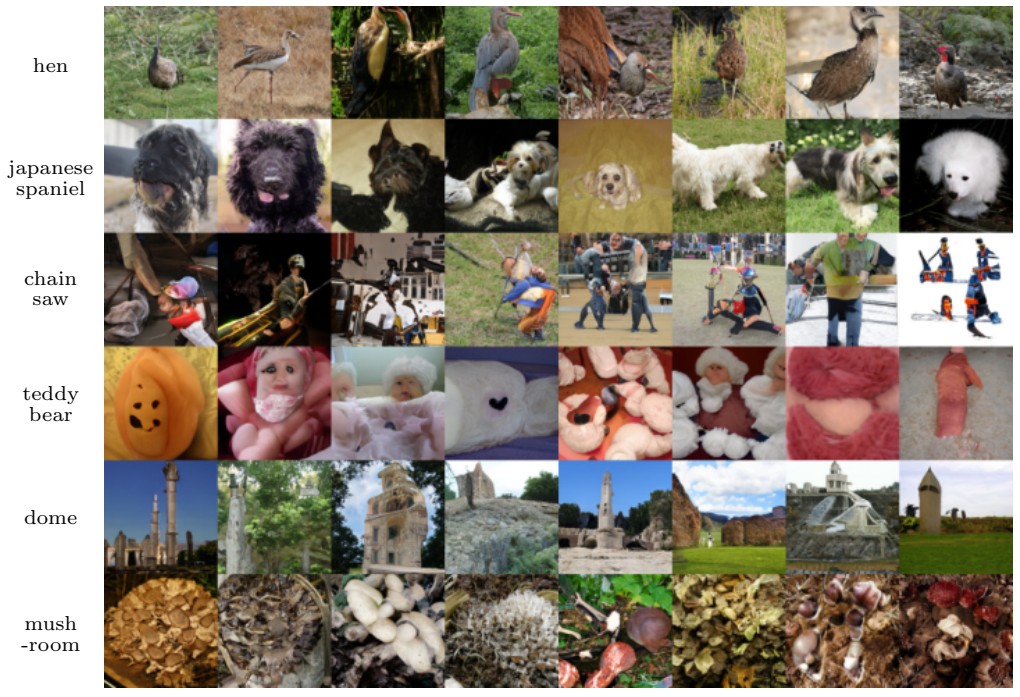

Figure A10: Generated images on ImageNet [47] dataset using ContraGAN [21] and the batch size of 256 (FID=25.16).

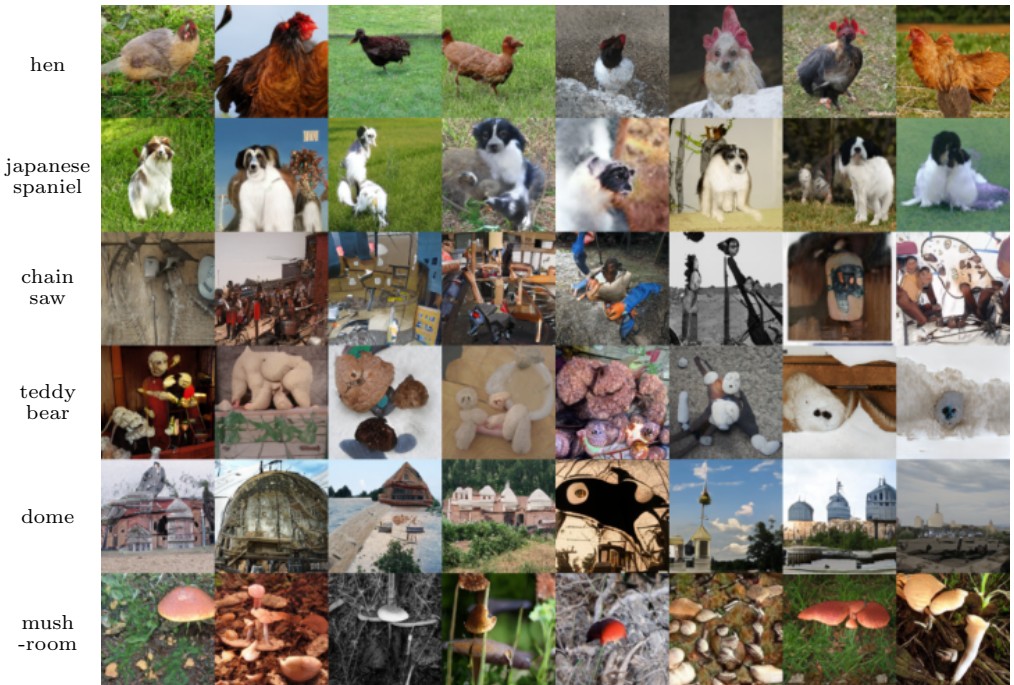

Figure A11: Generated images on ImageNet [47] dataset using BigGAN [16] and the batch size of 256 (FID=16.36).

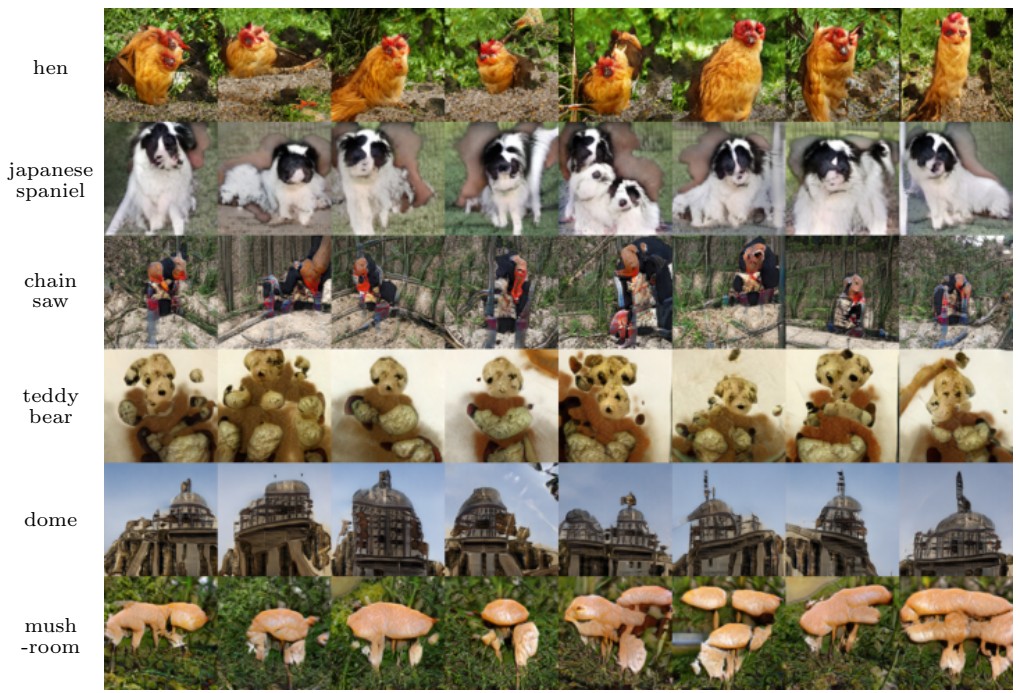

Figure A12: Generated images on ImageNet [47] dataset using ACGAN [10] and the batch size of 256 (FID=26.35).

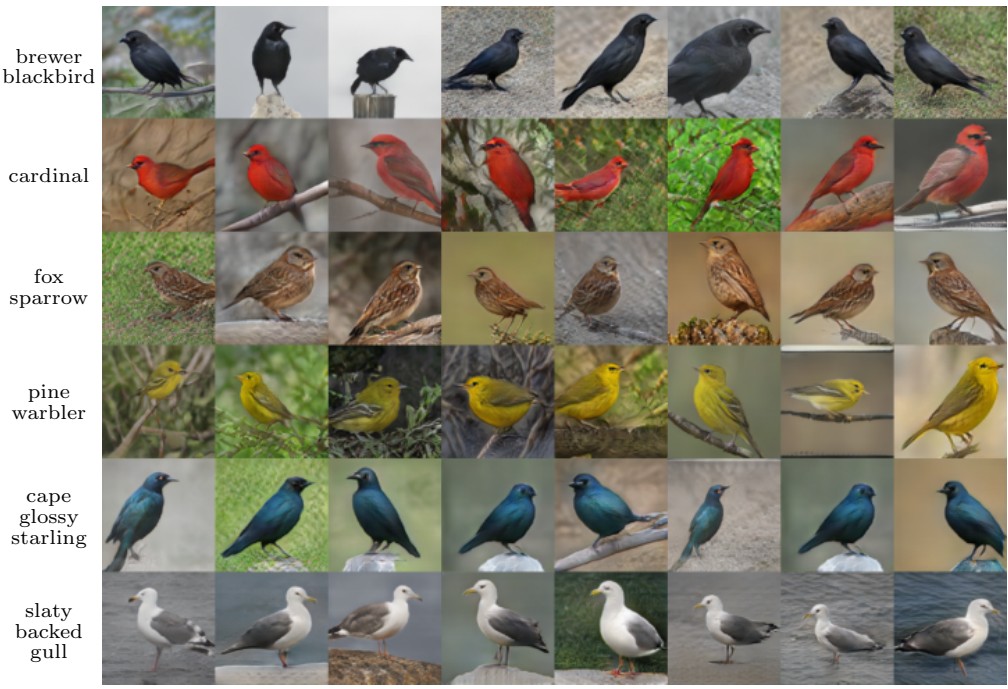

Figure A13: Generated images on CUB200 [46] dataset using ReACGAN (FID=14.67).

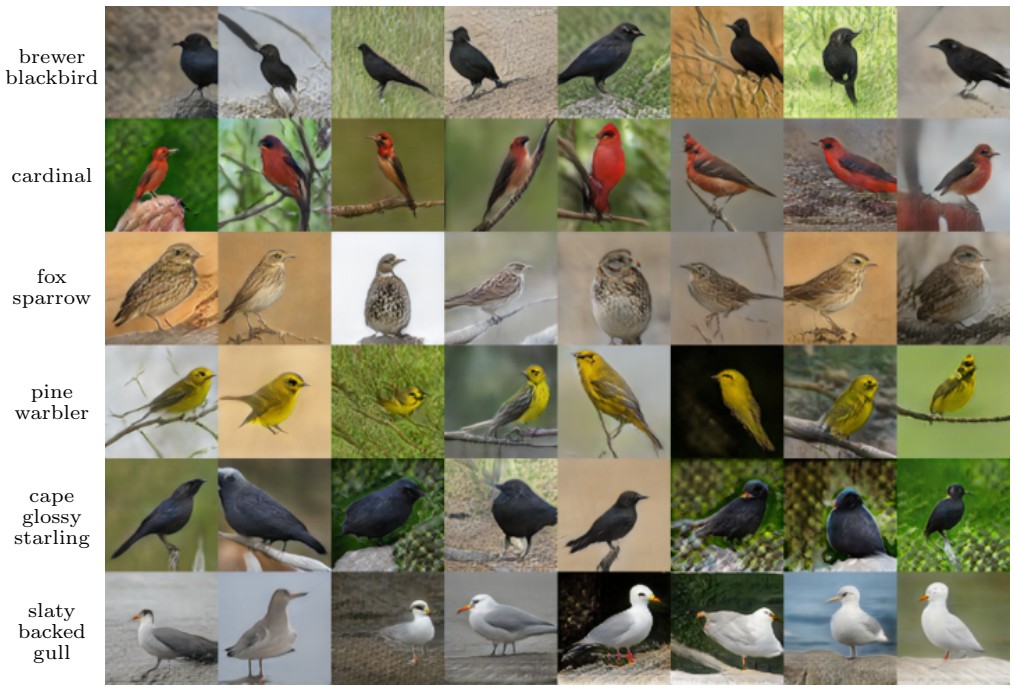

Figure A14: Generated images on CUB200 [46] dataset using ContraGAN [21] (FID=20.89).

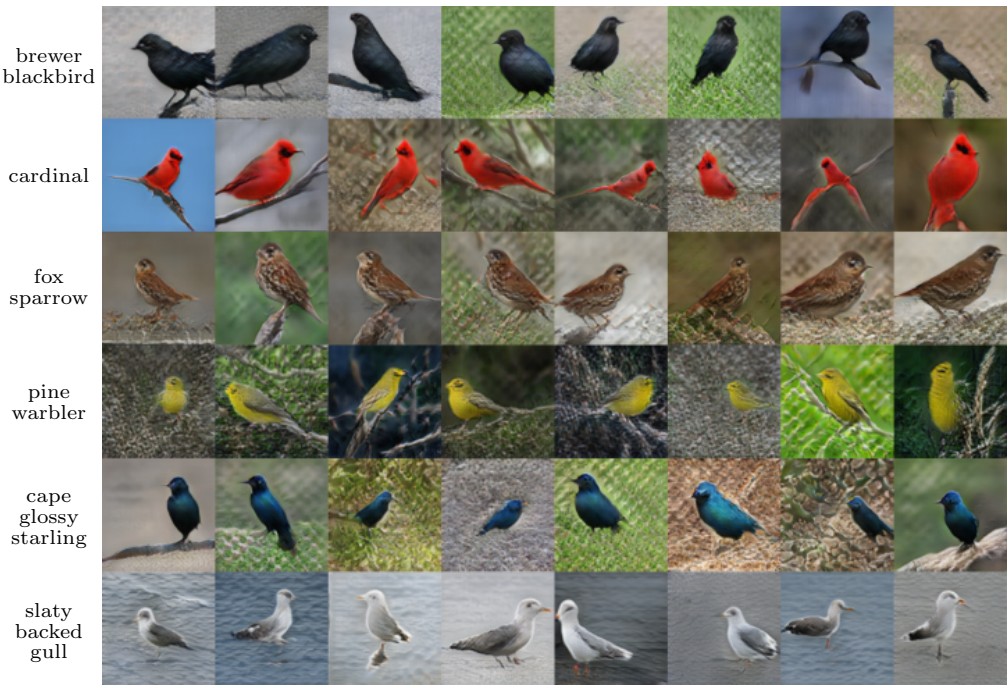

Figure A15: Generated images on CUB200 [46] dataset using BigGAN [16] (FID=17.80).

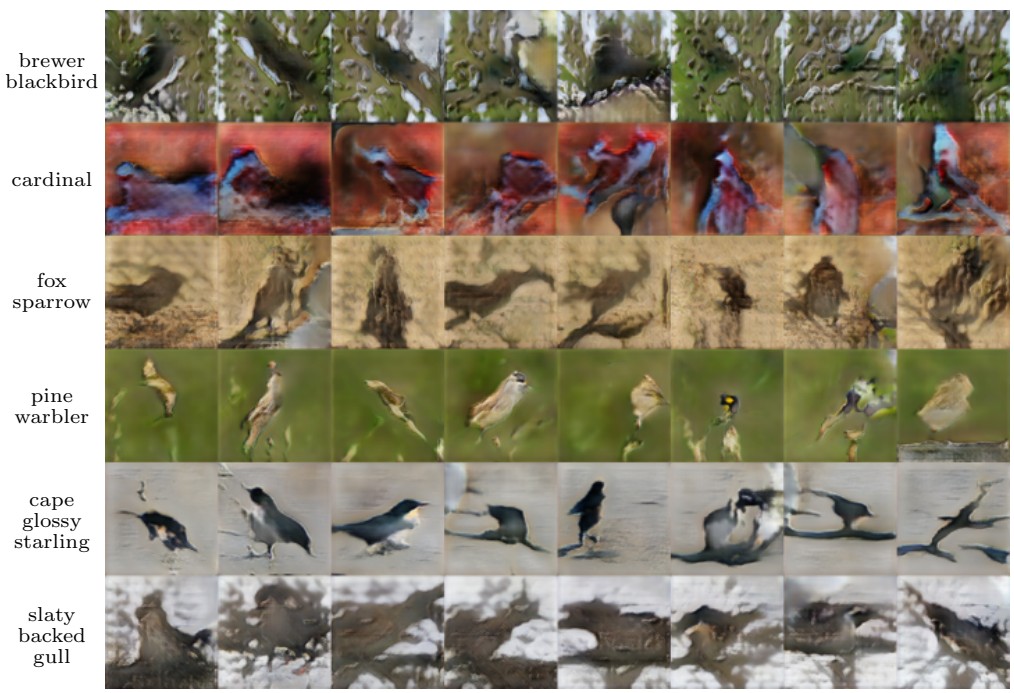

Figure A16: Generated images on CUB200 [46] dataset using ACGAN [10] (FID=61.29).

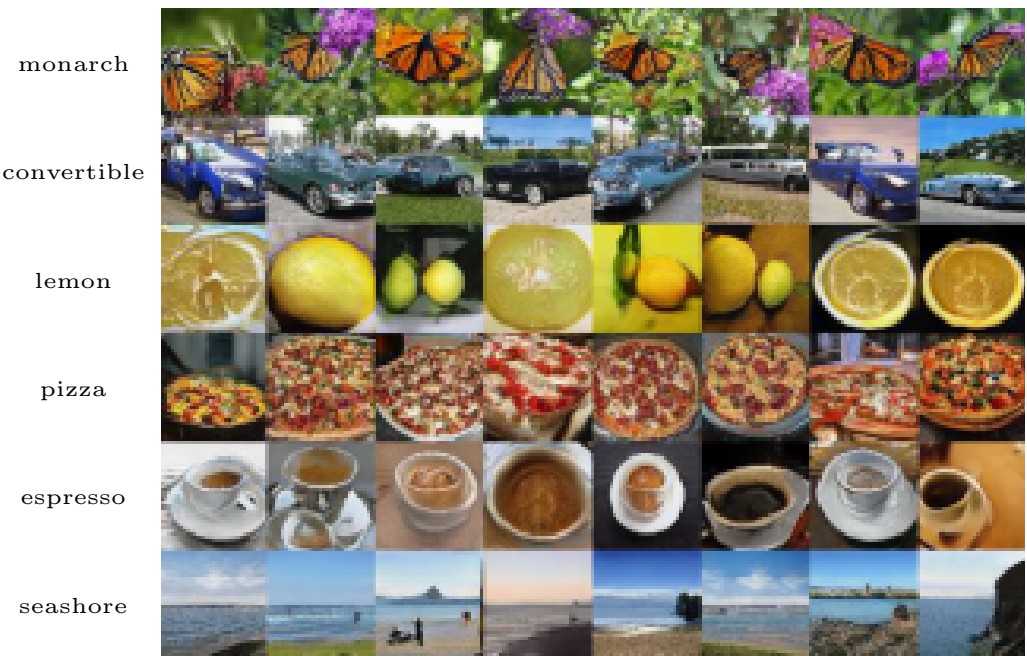

Figure A17: Generated images on Tiny-ImageNet [45] dataset using ReACGAN (FID=26.82).

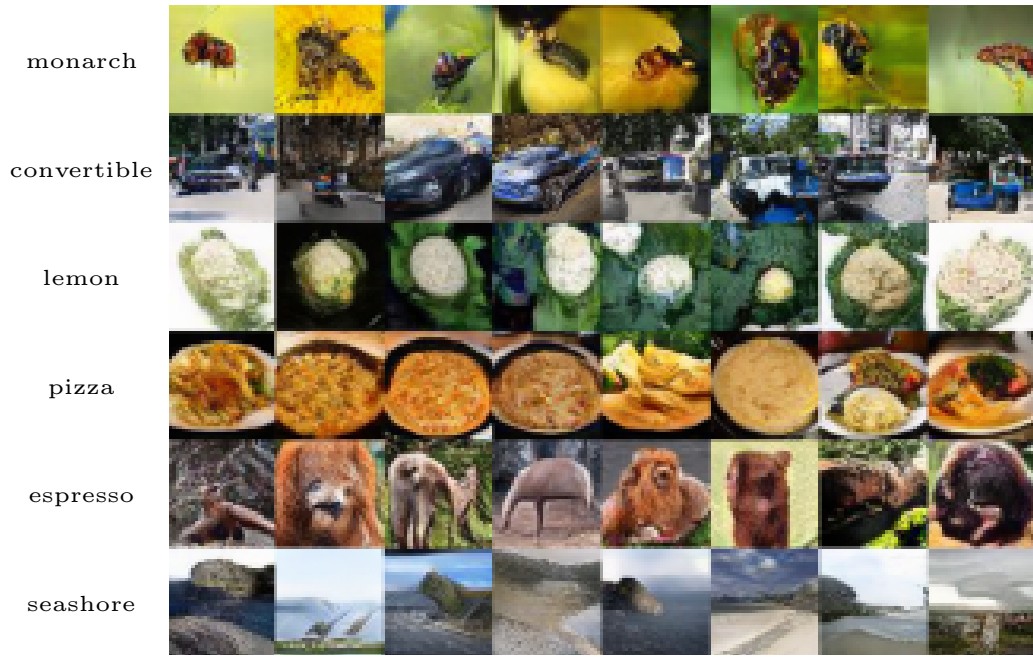

Figure A18: Generated images on Tiny-ImageNet [45] dataset using ContraGAN [21] (FID=28.41).

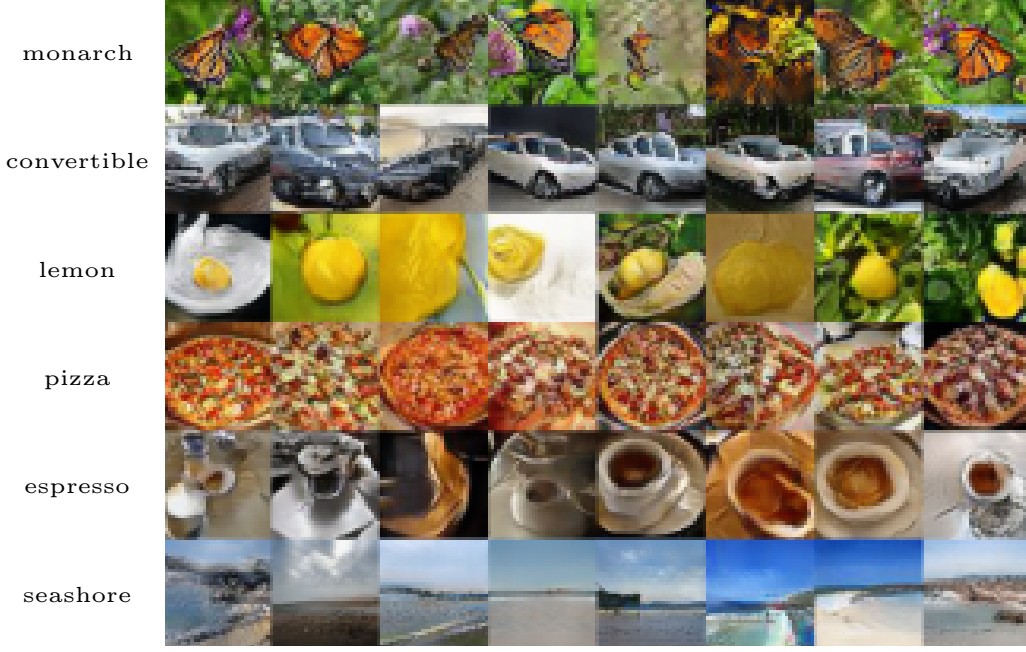

Figure A19: Generated images on Tiny-ImageNet [45] dataset using BigGAN [16] (FID=31.92).

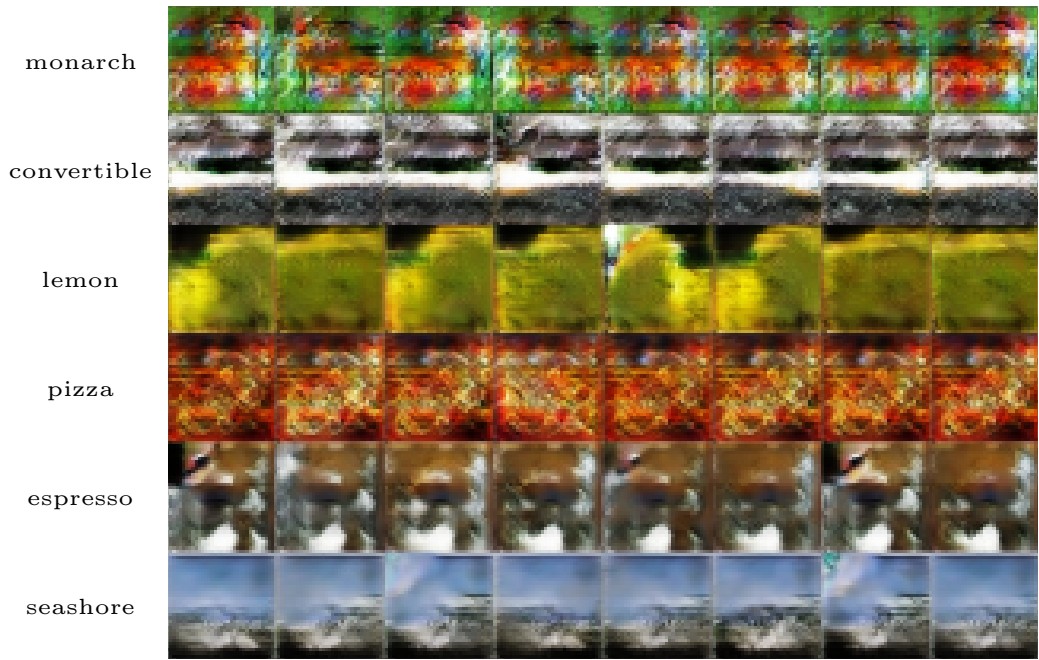

Figure A20: Generated images on Tiny-ImageNet [45] dataset using ACGAN [10] (FID=61.50).

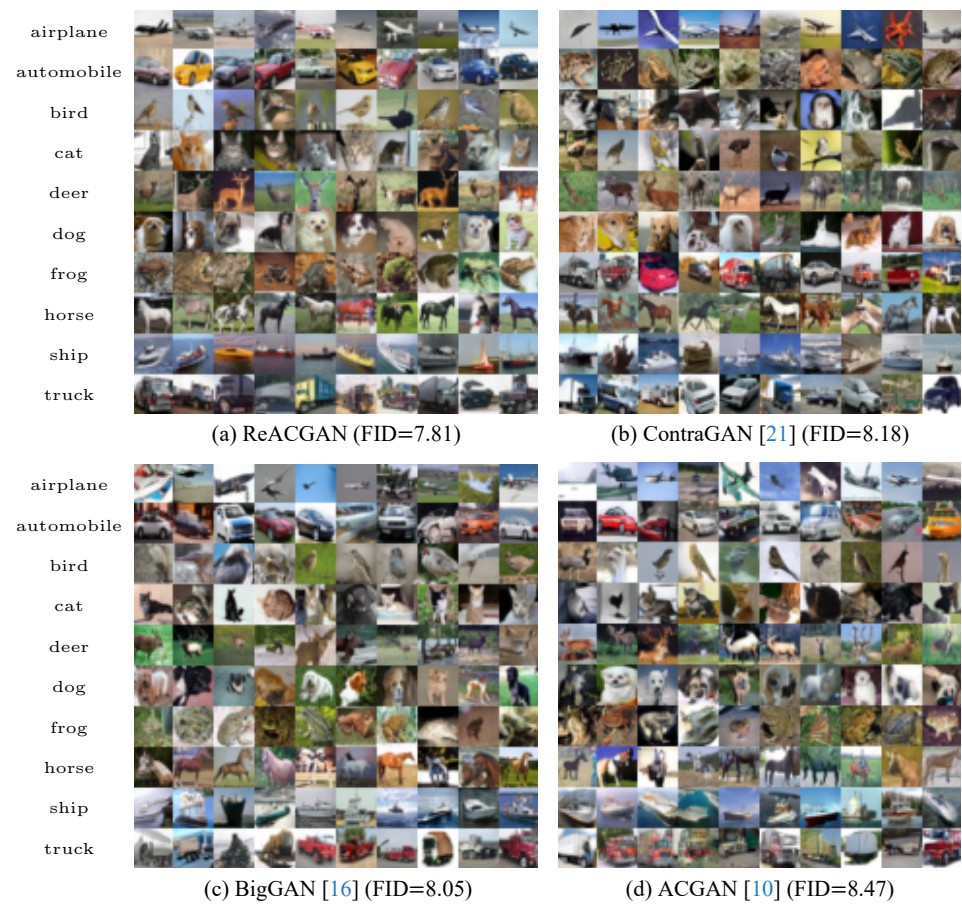

(a) ReACGAN (FID=7.81)

(b) ContraGAN [21] (FID=8.18)

(c) BigGAN [16] (FID=8.05)

(d) ACGAN [10] (FID=8.47)

Figure A21: Generated images on CIFAR10 [44] dataset.