# OpenReview forum: "Rebooting ACGAN: Auxiliary Classifier GANs with Stable Training"
_NeurIPS.cc/2021/Conference — NeurIPS 2021 Poster_

### Official Review · Reviewer_KmwA · 2021-07-07

**Rating:** 7
**Confidence:** 4

**Summary:**

This paper firstly identifies that the main reason for the GAN's training instability is gradient exploding, and normalize the discriminator's feature to avoid it. Then authors propose a D2D-CE loss to improve the generator model's generalization performance. The extensive experiments including ablation study show the effectiveness and superiority of the proposed method over baselines.

**Limitations And Societal Impact:**

See the section "Main Review"

**Main Review:**

Advantages:
   1. The presentation is clear and thus it is easy to follow the main idea.
   2. The proposed method is simple and well-motivated.
   3. Authors design extensive experiments in the main paper and supplementary material to evaluate the effectiveness of proposed methods, and the results are convincing.

Disadvantages:
   1. D2D-Loss needs to compute the similarity of Data-to-data in the same mini-batch and the similarity with their proxy, which may be time-consuming. It is better to report and compare the training time/device of the proposed method and baselines.
   2. I notice that the proposed method perform poorly on the Imagenet classification as Table A9 shows. One main reason is that the evaluation method can only evaluate the quality instead of the diversity of generated images. An alternative is training a classifier on the generated images and tests its accuracy on the real test set to evaluate both the quality and diversity of generated images. Specifically, the accuracies of the classifier may be unstable in the training procedure because of the randomness of generated images, and thus authors can report the best accuracies and std of each training procedure.
  3.  If the authors have enough time, it will be better to perform synthetic experiments in TAC-GAN, which is a simple way to evaluate whether the proposed methods can fit the true data distribution or not.

**Time Spent Reviewing:**

5

---

> ### Author Response · Authors · 2021-08-10
> **Reply to reviewer kmwA**
>
> We would like to thank the reviewer for providing thoughtful comments and constructive suggestions, which will be addressed thoroughly one-by-one in the following.
>
> \
> **C1. “It is better to report and compare the training time/device of the proposed method and baselines”**
>
> We investigate the time spent to train each model on Imagenet using eight Nvidia V100 GPUs. The batch size is set to 2048. We find that ReACGAN brings in a slight computational overhead and takes about 1.05~1.1 longer time than other algorithms. Specifically, BigGAN takes 17m 37s, ContraGAN 18m 24s, ReACGAN 18m 52s per 100 generator updates.
>
> \
> **C2. “training a classifier on the generated images and tests its accuracy on the real test set to evaluate both the quality and diversity of generated images”**
>
> Thanks for the constructive suggestion. However, due to the limited time, we could not provide the results of what was requested during the rebuttal period. We will try to conduct the experiment and add analysis in the final manuscript.
>
> \
> **C3. “If the authors have enough time, it will be better to perform synthetic experiments in TAC-GAN, which is a simple way to evaluate whether the proposed methods can fit the true data distribution or not.”**
>
> Thank you for the great suggestion. We conduct the 1D-MoG experiments in TAC-GAN using their official repository. We adopt the same losses, hyperparameters, architectures for both the discriminator and generator, and the target distribution with three Gaussian components, where the means and standard deviations are fixed to (0, 1), (3, 2), (6, 3), respectively. We also follow their evaluation protocol, where the maximum mean discrepancy (MMD) distances are computed between each distribution and the corresponding generated distribution. We compare ACGAN [18], BigGAN [4], and TACGAN [20] with our algorithm in Table R7. As a result, ReACGAN shows decent performance compared to ACGAN, and outperforms other algorithms when combined with TACGAN. For further comparison with TACGAN, we run the same experiments as ours using TACGAN on CIFAR10, Tiny ImageNet, and CUB200 and find that our method has indeed more strength to approximate the true distribution of the real dataset than the other approaches (Table R8).
>
> **Table R7**. MMD between each gaussian and marginal distributions and their corresponding generated distributions for ACGAN [18], BigGAN [4], TACGAN [20], and ReACGAN.
>
> |                           | Distance of class 0 |    1   |    2    | Distance of marginal |
> |---------------------------|:-------------------:|:------:|:-------:|:--------------------:|
> | ACGAN [18]                |        0.258        |  4.381 | 555.262 |        51.264        |
> | Projection Discriminator [19]                |        0.001        |  0.031 |  2.293  |         0.488        |
> | ReACGAN (ours)            |        0.213        | 25.544 |  63.482 |         0.752        |
> | ACGAN [18] + TAC [20]     |        0.055        |  0.046 |  -0.220 |         0.026        |
> | ReACGAN (ours) + TAC [20] |        0.065        | -0.016 |  -0.225 |        -0.173        |
>
> \
> **Table R8**. FID values (lower is better) of TACGAN and ReACGAN.
>
> |             | CIFAR10 [27] | Tiny Imagenet [30] | CUB200 [29] |
> |-------------|:------------:|:------------------:|:-----------:|
> | TACGAN [20] |     8.50     |        67.86       |    33.93    |
> | ReACGAN     |     **7.88**     |        **27.10**       |    **15.40**    |

---

> > ### Comment · Reviewer_KmwA · 2021-08-16
> > **Remaining Questions**
> >
> > Thank you very much for the response. I appreciate the effort that the authors put into addressing my questions but I still have some following questions.
> > 1. Thanks for reporting the training time. It would be nice to incorporate it into the final version.
> > 2. Thanks for your response, I understand the response time is too short to run such time-consuming experiments. I appreciate it if the authors can conduct the experiment and add analysis in the final manuscript because the performance of the classifier trained on the generated data is the key step for employing generation methods into practical applications.
> > 3. According to the results of Table R7, ReACGAN seems inferior to TACGAN in a simple synthetic experiment with a Ground-truth metric. Why is such a phenomenon, is ReACGAN hard to match the distributions with high overlap? On the other side, ReACGAN performs significantly better than TAC-GAN in some real datasets, e.g. TinyImagenet and CUB200. Is there any analysis about the performance gap between the two experiments (Synthetic data and Real data)?

---

> > > ### Author Response · Authors · 2021-08-18
> > > **Reply to reviewer's question 3**
> > >
> > > Thank you so much for your interest and the thoughtful response. We will answer the remaining questions below.
> > >
> > > Before starting, we would like to note two facts. (1) ReACGAN is an evolved ACGAN whose conditioning loss is changed to D2D-CE from CE. (2) ReACGAN is also able to benefit from the twin auxiliary classifier (TAC) in a particular case (learn conditional distributions with significant overlap. We denote the overlapped conditional distributions as overlapped distributions for a convenient).
> > >
> > > \
> > > **Q1. ReACGAN hard to match the distributions with high overlap?**
> > >
> > > ReACGAN seems to have difficulty in approximating the overlapped distribution. We speculate that this is because ReACGAN imposes perfect separability for class conditioning like ACGAN. Please refer to the line 8-9 in the abstract of TACGAN paper [20].
> > >
> > > ```
> > > line 8-9
> > > AC-GAN imposes perfect separability, which is disadvantageous when the supports of the class distributions have significant overlap.
> > > ```
> > >
> > > Although ReACGAN has a room for improvement through a proper hyperparameter selection, ReACGAN can be reinforced with the twin auxiliary classifier (TAC) to approximate the overlapped distribution more precisely, as shown in Table R7 (3rd and 5th rows).
> > >
> > > \
> > > **Q2. On the other side, ReACGAN performs significantly better than TAC-GAN in some real datasets, e.g., TinyImagenet and CUB200. Is there any analysis about the performance gap between the two experiments?**
> > >
> > > While TACGAN can successfully approximate the overlapped conditional distributions on the synthetic dataset, some works [Ref 1, Ref 2, TableR8] have reported that training TACGAN on the real dataset is still unstable. As our paper demonstrated, classifier-based GANs that use a cross-entropy loss can suffer from an early training collapse problem when the number of classes is large. Thus, we speculate that TACGAN also undergoes the same early collapse problem, and the normalization trick can resolve the unstable training issue.
> > >
> > > Table R9. FID values of ACGAN [18], ReACGAN, TACGAN (ACGAN + TAC) [20], and ReACGAN + TAC [20] on Tiny_Imagenet [30] dataset. Since we conducted all experiments regarding ReACGAN + TAC once, we hope selecting proper hyperparameters will ensure better generation performance on real datasets.
> > >
> > > |                                              | Tiny Imagenet [30] |
> > > |----------------------------------------------|:------------------:|
> > > | ACGAN [18]                                   |        96.04       |
> > > | ReACGAN                                      |        27.10       |
> > > | ACGAN [18] + TAC [20]                        |        67.86       |
> > > | ACGAN [18] + TAC [20] + Normalization (ours) |        29.07       |
> > > | ReACGAN (ours) + TAC [20]                    |        **26.14**       |
> > >
> > > \
> > > Table R9 supports our main claim that the normalization trick can effectively prevent the early training collapse problem of AC/TACGAN. In addition, TAC + ReACGAN shows superior results than the ACGAN + TAC + Normalization model. We think this is because considering data-to-data relationships gives useful supervisory signals for image generation to both the generator and discriminator, as demonstrated in our main paper. We expect that incorporating TAC (twin auxiliary classifier) or ADC (auxiliary discriminative classifier) [Ref 2] into ReACGAN further improves the quality and diversity of generated images. We will try to conduct more experiments using TACGAN and reflect the results in our final manuscript.
> > >
> > > [Ref 1] Han, L., Stathopoulos, A., Xue, T., & Metaxas, D.N. (2020). Unbiased Auxiliary Classifier GANs with MINE. ArXiv, abs/2006.07567.
> > >
> > > [Ref 2] Hou, L., Cao, Q., Shen, H., & Cheng, X. (2021). cGANs with Auxiliary Discriminative Classifier. ArXiv, abs/2107.10060.

---

> > > > ### Comment · Reviewer_KmwA · 2021-08-21
> > > > **Re: Reply**
> > > >
> > > > Thanks for your response which address my concerns. I think incorporating the discussions and experimental results during the rebuttal period to the final draft could significantly improve the clarity significance, and the limitation of the proposed method be clear.

---

### Official Review · Reviewer_JFMx · 2021-07-11

**Rating:** 6
**Confidence:** 4

**Summary:**

The paper proposes approaches to reboot GANs with auxiliary classifier: a) normalizing the classifier layer and b) a data-to-data cross-entropy loss that encourages the auxiliary classifier to learn different class embeddings for different instances. Experiments demonstrate state-of-the-art class-conditional generation performance on CIFAR10, CUB200, and simplified ImageNet datasets with the specified backbone and better performance when working with CR and DiffAugment.

**Limitations And Societal Impact:**

Good

**Main Review:**

Strengths:
- The proposed techniques are well-motivated, sound novel and reasonable.
- With the proposed techniques, the paper successfully brings auxiliary classifier, an alternative that was deemed to be surpassed by the projection-based approach, into state-of-the-art cGANs, which may invoke new thoughts about auxiliary classifier in the domain.

Weaknesses:
- The experiments are limited to BigGAN related backbones. The readers would also be interested about whether the proposed techniques can help StyleGANs for conditional generation.
- I think deeper analysis is needed to fully understand the proposed techniques. I am curious about why the proposed normalization technique can help even when Spectral Normalization is used. Is the last classifier layer using SN? I think that the gradient issue as the authors discovered may not be the root cause. Is it related to discriminator overfitting? Can we alleviate the issue by gradient clipping or lowering the weight of the auxiliary cross-entropy loss?

**Time Spent Reviewing:**

4

---

> ### Author Response · Authors · 2021-08-10
> **Reply to reviewer JFMx**
>
> We thank the reviewer for the valuable comments, especially for acknowledging the novelty of the paper that opens up a new design space of GANs based on an auxiliary classifier. Here, we try to answer the reviewer’s concerns one by one.
>
> \
> **C1. "The readers would also be interested about whether the proposed techniques can help StyleGANs for conditional generation."**
>
> Thank you for your good suggestion. We tried conditional image generation experiments using D2D-CE and StyleGANv2 [7] backbone, but we failed to train the StyleGAN2 model. We speculate that training StyleGANv2 needs careful hyperparameter tuning for path length, lazy, and R1 regularizations. The authors of OmniGAN [24] also reported that they could not train StyleGANv2 using the omni loss they proposed. Although we fail to bring meaningful numbers during the rebuttal period, we promise to continue conducting StyleGANv2 experiments and report results regardless of the success of the experiments in our main paper.
>
> As an alternative, we conduct additional experiments using two popular backbone architectures (SNCNN [3] and SNResnet [3]) below. The results (Table R2, 3, 4) demonstrate that ReACGAN gives better synthesis results than ACGAN [18], Projection discriminator [19], and ContraGAN [22] across various architectures.
>
> \
> **Table R2**. Results of conditional image generation experiments using CIFAR10 [27] and SNCNN backbone [3].
>
> | Backbone: SNCNN [3] | ACGAN [18] | Projection Discriminator [19] | ContraGAN [22] | ReACGAN |
> |-------------------|:----------:|:-----------------------------:|:--------------:|:-------:|
> | IS [31]           |    **8.42**    |              7.76             |      8.03      |   8.25  |
> | FID [32]          |    19.75   |             21.35             |      22.24     |  **18.81**  |
> | F_0.125 [33]      |    **0.972**   |             0.970             |      0.962     |  0.967  |
> | F_8 [33]          |    0.960   |             0.948             |      0.902     |  **0.966**  |
>
> \
> **Table R3**. Results of conditional image generation experiments using CIFAR10 [27] and SNResnet backbone [3].
>
> | Backbone: SNResnet [3] | ACGAN [18] | Projection Discriminator [19] | ContraGAN [22] | ReACGAN |
> |----------------------|:----------:|:-----------------------------:|:--------------:|:-------:|
> | IS [31]              |    8.91    |              8.84             |      8.917     |   **9.07**  |
> | FID [32]             |    13.87   |             13.75             |      13.52     |  **12.58**  |
> | F_0.125 [33]         |    0.984   |             0.983             |      0.984     |  **0.986**  |
> | F_8 [33]             |    0.963   |             0.972             |      0.973     |  **0.980**  |
>
>
> \
> **Table R4**. Results of conditional image generation experiments using Tiny Imagenet [30] and SNResnet backbone [3].
>
> | Backbone: SNResnet [3] | ACGAN [18] | Projection Discriminator [19] | ContraGAN [22] | ReACGAN |
> |----------------------|:----------:|:-----------------------------:|:--------------:|:-------:|
> | IS [31]              |    6.45    |              9.13             |      10.04     |  **10.53**  |
> | FID [32]             |   103.13   |             48.42             |      43.30     |  **41.34**  |
> | F_0.125 [33]         |    0.676   |             0.905             |      **0.932**     |  0.931  |
> | F_8 [33]             |    0.432   |             0.727             |      0.759     |  **0.770**  |
>
> \
> **C2. Further analysis about the early training collapse.**
>
> **[why the proposed normalization technique can help even when Spectral Normalization is used. Is the last classifier layer using SN?]**
>
> Yes, the last layer of the classifier uses spectral normalization [3]. We state this in lines 219--220 of our main paper.
>
> Suppose we apply spectral normalization to the weight in the classifier. Now we try to understand the gradient of the cross-entropy (CE) loss w.r.t naive weight vector w_{k} to explain why training ACGAN with SN can collapse at the early training phase. CE loss will be calculated with a feature F(x) and w_{sn}. Thus, we can replace w_{k} in Eq.4 with w_{sn, k}, and this means that the gradient of CE w.r.t w_{sn, k} is also related to the norm of the feature vector. Since the gradient of w_{sn, k } w.r.t w_{k} will be multiplied by the gradient of CE w.r.t w_{sn, k} for backpropagation, we now know that the gradient of CE loss w.r.t w_{k} is also related to the norm of a feature vector. For this reason, ACGAN with spectral normalization also suffers from the early training collapse (please refer to Figure 2 in our main paper and Figure C.2 in appendix).
>
> However, if we apply normalization on the feature map F(x), the training collapse disappears since the norm of the feature vector F(x) is restricted under 1. To quantitatively show the gradient exploding phenomenon caused by the large norm of the feature map, we conduct an extra experiment using spectral normalized ACGAN and Tiny Imagenet dataset. The result is summarized in Table R5. The results imply that the early training collapse is not caused by the discriminator's overfitting since the probability values of the classifier are still low. Also, we can identify that the stretching out features and low probability values can cause the classifier’s gradient to explode. We will add the above explanation in our main paper.
>
> **Table R5**.  Average norms of feature vectors (F(x)) and gradients in the ACGAN classifier [18] and the average value of target probabilities (p_{*, k} in the main paper). We apply spectral normalization for the discriminator of ACGAN and do not apply the feature normalization trick we proposed.
>
> | Backbone: ACGAN [18]       | 1K iteration |    2K   |    3K   |    4K   | 5K      |
> |----------------------------|:------------:|:-------:|:-------:|:-------:|---------|
> | FID [32]                   |    153.421   | 126.691 | 112.142 | 106.410 | 107.973 |
> | Average norm of features   |    23.759    |  28.951 |  42.963 |  62.191 | 95.851  |
> | Average value of target probabilities |     0.013    |  0.029  |  0.086  |  0.130  | 0.245   |
> | Average norm of gradients  |     0.055    |  0.072  |  0.097  |  0.123  | 0.241   |
>
> \
> **C3. Can we alleviate the issue by gradient clipping or lowering the weight of the auxiliary cross-entropy loss?**
>
> We conducted more experiments to identify the effectiveness of the lowering technique, feature clipping technique, and the normalization trick. The results are summarized in Table R6. Lowering the weight of the cross-entropy loss shows gradual improvement in image generation performance as the lowering coefficient decreases. In addition, explicitly clipping the feature vector (we set the interval to (-0.01, 0.01)) can help ACGAN training further. Notably, the proposed normalization technique outperforms the other methods (lowering and clipping).
>
> We would like to emphasize that these experimental results support our main idea that large gradients from the classifier can break adversarial dynamics (line 109--110), and restricting the norm of feature maps is an effective way to prevent the early training collapse problem (line 114--115).
>
> **Table R6**. Experimental results of investigating the effectiveness of lowering the weight of the ACGAN loss, clipping the feature vector, and normalizing the feature map for ACGAN stabilizing.
>
> |          | Unconditional |  coefficient=1.0 |  0.75 |   0.5  |  0.25 |  0.1  | Clipping | Normalization |
> |:--------:|:-------------:|:------:|:-----:|:------:|:-----:|:-----:|:--------:|:-------------:|
> | IS [31]  |      7.23     |  5.21  |  7.22 |  6.49  |  9.34 | 10.09 |   7.84   |     **11.27**     |
> | FID [32] |     71.53     | 105.66 | 99.18 | 103.90 | 66.02 | 45.43 |   59.36  |     **38.56**     |

---

> > ### Comment · Reviewer_JFMx · 2021-08-25
> > **Reply**
> >
> > Thanks very much for the response. I like the added analysis, which would make the rationality of the proposed method stronger.

---

### Official Review · Reviewer_mphj · 2021-07-17

**Rating:** 6
**Confidence:** 3

**Summary:**

This paper proposes ReACGAN which is an improved version of ACGAN. The proposed method addresses the gradient exploding of ACGAN in the early training stage by normalizing the feature embeddings onto a unit hypersphere. The authors propose D2D-CE, which is analogous to contrastive loss, to effectively utilize relational information in the dataset. In addition, the authors introduce thresholds to focus on pushing and pulling hard negative and positive samples, further enhancing the conditional generation capability of ACGAN.

**Limitations And Societal Impact:**

Yes

**Main Review:**

This paper is well-written and the authors well investigated the problem of ACGAN and the  proposed method both theoretically and empirically. However, my concern is the novelty of the proposed method because most of the techniques here have been explored in other works (D2D-CE is similar to [1], and the motivation of the thresholds is similar to [2].).

[1] ContraGAN: Contrastive Learning for Conditional Image Generation, NeurIPS 2020.

[2] Focal Loss for Dense Object Detection, ICCV 2017.


I have a minor question: MHGAN paper reported IS and FID on Imagenet dataset. Was there any reason not to include this result in Table 2 of the paper?


Despite the concerns, the authors showed strong experimental results, and I look forward to acceptance.


**Time Spent Reviewing:**

6 hours

---

> ### Author Response · Authors · 2021-08-10
> **Reply to reviewer mphj**
>
> We thank you for your thoughtful comments on our study. In this rebuttal, we try to answer some concerns raised by the reviewer carefully.
>
> **C1. The novelty of the proposed method because most of the techniques here have been explored in other works (D2D-CE is similar to [1], and the motivation of the thresholds is similar to [2]).**
>
> **[Difference between 2C loss and D2D-CE]**
>
> We would like to highlight that the differences between 2C loss and D2D-CE can be summarized into two folds: (1) D2D-CE and 2C loss stem from different motivations, and (2) the two losses operate differently in the perspective of hard sample mining.
>
> In the ContraGAN paper, the authors point out that considering data-to-data relationships gives useful relational information for GAN training, and they start from NT-Xent loss (data-to-data relationships) to propose 2C loss (data-to-data + data-to-class relationships) for conditional image generation. On the other hand, the proposed D2D-CE loss is devised for the prevention of the early training collapse of ACGAN (data-to-class). We analytically and experimentally demonstrate that the proposed normalization technique is effective for preventing the early collapse. On this foundation, we add the idea of considering data-to-data relationships. Thus, the motivations of D2D-CE and 2C loss are different from each other.
>
> Besides, as stated in lines 177--189 in our main paper and attached Appendix E, D2D-CE and 2C loss of ContraGAN operate differently in the perspective of hard sample mining. Specifically, 2C loss has multiple similarities between an anchor and its positive samples in the numerator. According to the authors’ explanation [22], this formulation is adapted to alleviate the false negative repulsion force reported by [A25, A26, A27]. However, as can be seen in appendix E of our supplement, the similarities between an anchor and the false negatives in the numerator of 2C loss can cause undesirable easy positive mining. Moreover, the false negative terms in the denominator of 2C loss can attenuate a negative repulsion force. These characteristics can cause the embedding features of the discriminator to be clustered in an unintended way and induce a class confusion problem reported in Appendix E (line 101--104 in the attached supplement) and Appendix H. As a result, Top-1 classification accuracy of generated images is only 2.866% for ContraGAN while ReACGAN is relatively free from the class confusion problem.
>
> For these reasons, We would like to argue that D2D-CE is a novel objective that can reduce training instability caused by softmax cross-entropy in ACGAN and effectively consider relational information between data points.
>
> **[Motivation of the threshold is similar to Focal loss]**
>
> The motivation of the threshold used in our paper is to suppress gradients from easy samples and seems to be similar to the motivation of Focal loss. However, we want to highlight that the mechanism and positive effects of the threshold technique used in our paper are different from those of Focal loss.
>
> Specifically, D2D-CE utilizes ReLU activation to suppress both easy negative (low similarity) and easy positive samples (high similarity), while Focal loss represses only easy-negative samples for enhanced training (please refer to Figure 1 and section 3.3 of our paper for more details). Also, thresholding positive similarity can roll as a controller on fidelity and diversity trade-off. If the positive threshold value is high, the generator tries to generate samples whose similarities between corresponding proxies are also high. As a result, the generator is likely to generate high-fidelity (well-classifiable) but relatively low diverse images.
>
> We found this trade-off when we trained ReACGAN on the Imagenet dataset. Due to the limited rebuttal time, we promise to add a more thorough analysis of the fidelity and diversity trade-off caused by the threshold technique in the final manuscript.
>
> \
> **C2. MHGAN paper reported IS and FID on Imagenet dataset. Was there any reason not to include this result in Table 2 of the paper?**
>
> The experiment of MHGAN [25] on Imagenet in the original paper is conducted under large batch circumstances (batch size of 1024), whereas Table 2 of the paper compares the results of experiments that use a batch size of 256. Therefore, we did not include the numbers in MHGAN paper to avoid confusion.
>
> During the rebuttal period, we tried to train MHGAN using a batch size of 256. Due to the lack of exact configuration for Imagenet training, we failed to train MHGAN using the official implementation. Thus we borrowed the main components of MHGAN from the official repository and trained MHGAN on StudioGAN [39] using a batch size of 256. With the result of MHGAN, we report additional experimental results of ReACGAN on Imagenet as below:
>
> **Table R1.** Additional experimental results on Imagenet dataset. MHGAN [39] is trained using a batch size of 256. MHGAN [25] shows the numbers reported in the original MHGAN paper (conducted using a batch size of 1024). Except for the experiments using MHGAN, all experiments are conducted using a batch size of 2048.
>
> |              | MHGAN [39] | MHGAN [25] |  BigGAN [4] | BigGAN [24] | BigGAN [39] | ReACGAN (ours) |
> |:------------:|:----------:|:----------:|:-----------:|:-----------:|:-----------:|:--------------:|
> | IS [31]      |    11.02   |    61.98   | 99.31 +-2.1 |    **104.57**   |    99.71    |      92.74     |
> | FID [32]     |    74.23   |    13.27   |  8.51+-0.32 |     9.18    |     **7.89**    |      8.23      |
> | F_0.125 [33] |    0.720   |            |      -      |      -      |    0.985    |      **0.991**     |
> | F_8 [33]     |    0.552   |            |      -      |      -      |    0.989    |      **0.990**     |
>
> Please note that the training setup and architecture of BigGAN have been hugely explored while ReACGAN on Imagenet 2048 batch experiment is relatively less searched (only two runs). Although ReACGAN shows slightly higher FID compared to BigGAN [39], F_{0.125} (precision) and F_{8} (recall) values show that ReACGAN can generate more precise images without losing diversity.

---

### Official Review · Reviewer_1hzt · 2021-07-19

**Rating:** 7
**Confidence:** 4

**Summary:**

This paper proposes to revisit the AC-GAN framework, which usually suffers from training instability, mode collapse and lack of diversity. The authors identify two limitations and propose solutions. First, the classifier causes exploding gradients which is addressed by normalizing feature embeddings. Second, the authors replace the softmax classifier by a contrastive classifier. The experimental results show consistent improvements in the common metrics.

**Limitations And Societal Impact:**

There is not any section about limitations, but the authors have included a section in the supplementary describing potential misuses of GANs and total training time.

**Main Review:**

Originality:
While AC-GAN, L2 normalized features and contrastive learning are certainly not novel and well-known, the study of the AC-GAN gradient problem is somewhat novel. The authors propose an effective method to address the problem and a modified contrastive loss showing certain improvement over ContraGAN. It can be seen as an effective combination of AC-GAN and ContraGAN.

Quality:
The submission is technically sound, with extensive theoretical and experimental analysis. The authors explained in detail the relation and differences with closely related works.

Clarity:
The submission writing is very clear and well organized.

Significance:
The results are significant, especially in Imagenet, which is the most challenging setting. The conclusions are solid and provide valuable insight on the interplay between classifier and GAN.

**Time Spent Reviewing:**

5

---

> ### Author Response · Authors · 2021-08-10
> **Reply to reviewer 1hzt**
>
> We would like to thank the reviewer for recommending acceptance. We are pleased that the reviewer likes the study of the AC-GAN’s gradient problem and our approach to address it. Our method is indeed simple, yet effective in most experimental setups, especially at the most challenging setting, Imagenet. We conducted more experiments regarding a large-scale image generation experiment on Imagenet (Table R1 in response to reviewer mphj), robustness against architectural selection (Table R2~R4 in response to reviewer JFMx), and TACGAN experiments (Table R7-R8 in response to KmwA) during the rebuttal period for understanding ReACGAN further.

---

### Decision · Program_Chairs · 2021-09-28

**Decision:**

Accept (Poster)

**Comment:**

 This paper proposes two improvements to address the low diversity problem of auxiliary classifiers GANs. First, the classifier is projected to a unit hypersphere to avoid early training collapse caused by gradient exploding. Second, a data to data cross entropy loss, which is similar to the contrastive gan loss, is used to better explore class information. Though the proposed method is rather heuristic, it shows impressive results in conditional image generation on real natural images. The reviewers agree that the proposed method is technically sound, novel, and the paper is well written and organized. However, as pointed out by reviewer KmwA, the proposed method does not mitigate the theoretical problem of ACGAN when the class distributions have support overlaps, which is also verified on the additional experiments on the simulated data.  I would recommend acceptance of this paper given its novelty and impressive performance, and I highly suggest the authors add simulations as done in the TAC paper and report the results of the combination of TAC in their real experiments (with proper discussions), as suggested by reviewer KmwA.

**Consistency Experiment:**

NeurIPS has a long history of experimentation. In 2014, NeurIPS ran an experiment in which 10% of submissions were reviewed by two independent committees to quantify the randomness in the review process. This year, we repeated a variant of this experiment to see how the quality of the review process has changed over time.  This paper was part of the experiment and was therefore assigned to two committees (consisting of reviewers, an Area Chair, and a Senior Area Chair) that reached independent decisions.  If both committees made the same recommendation, this recommendation was followed. If a single committee recommended acceptance, the paper was accepted (with the exception of a few cases in which the other committee identified what we considered a fatal flaw, e.g., an error in a key result).

This copy’s committee reached the following decision: **Accept (Poster)**

The other committee assigned to the paper recommended **Reject**.  You can find the other set of reviews, along with any follow up discussion with the authors here:
https://openreview.net/forum?id=r7UC-b67YkO